# An observational network of ground surface temperature under different landcover types on northeastern Qinghai-Tibet Plateau

Raul-David Șerban[1, 2, 3 *], Huijun Jin[3, 4 *], Mihaela Șerban[5], Giacomo Bertoldi[2], Dongliang Luo[3], Qingfeng Wang[3], Qiang Ma[3], Ruixia He[3], Xiaoying Jin[4], Xinze Li[3, 6, 7], Jianjun Tang[4], and Hongwei Wang[3, 4]

[1] Faculty of Agricultural, Environmental and Food Sciences, Free University of Bozen-Bolzano, Bolzano 39100, Italy;
[2] Institute for Alpine Environment, Eurac Research, Bolzano 39100, Italy;
[3] National Key Laboratory of Cryosphere Science and Frozen Soils Engineering, Northwest Institute of Eco-Environment and Resources, Chinese Academy of Sciences, Lanzhou 730000, China;
[4] School of Civil Engineering and Transportation, Permafrost Institute, and China-Russia Joint Laboratory of Cold Regions Engineering and Environment, Northeast Forestry University, Harbin 150090, China;
[5] Applied Geomorphology and Interdisciplinary Research Centre, Department of Geography, West University of Timișoara, Timișoara 300223, Romania;
[6] College of Resources and Environment, University of Chinese Academy of Sciences, Beijing 100049, China, and;
[7] College of Engineering, China University of Petroleum-Beijing at Karamayi 834000, China

*Correspondence to*: Huijun Jin (hjjin@nefu.edu.cn) and Raul-David Șerban (RaulDavid.Serban@unibz.it)

**Abstract.**

Ground surface temperature (GST), measured at approximately 5 cm in depth, is a key controlling parameter for subsurface biophysical processes at the land-atmosphere boundary. This work presents a valuable dataset of GST observations at various spatial scales in the Headwater Area of the Yellow River (HAYR), a representative area of high-plateau permafrost on northeastern Qinghai-Tibet Plateau (QTP). GST was measured every three hours using 72 iButton temperature loggers (DS1922L) at 39 sites from 2019 to 2020. At each site, GST was recorded in two plots at distances from 2 to 16 m under similar and different landcover conditions (steppe, meadow, swamp meadow, and bare ground). These sensors proved their reliability in harsh environments because there were only 165 biased measurements from a total of 210,816. A high significant correlation ($> 0.96$, $p < 0.001$) was observed between plots, with a mean absolute error (MAE) of 0.2 to 1.2 °C. The daily intra-plot differences in GST were mainly $< 2$ °C for sites with similar landcover in both plots and $> 2$ °C when GST of bare ground was compared to that of sites with vegetation. From autumn to spring, the differences in GST could increase to 4–5 °C for up to 15 days. The values of the frost number (FN) were quite similar between the plots with Differences in FN $< 0.05$ for most of the sites. This dataset complements the sparse observations of GST on the QTP and helps to identify the permafrost distribution and degradation at high resolution and to validate and calibrate the permafrost distribution models. The datasets are openly available in the National Tibetan Plateau/Third Pole Environment Data Center (https://dx.doi.org/10.11888/Cryos.tpdc.272945, Șerban and Jin, 2022).

**Keywords**: Ground surface temperature (GST), permafrost, fine-scale, landcover, Qinghai-Tibet Plateau (QTP)

## 1 Introduction

Known as the Roof of the World, or the core of the Third Pole, the Qinghai-Tibet Plateau (QTP) is the largest and highest plateau in the world due to its complex terrains and an average elevation of 4000 m a. s. l. (Li et al., 2022). The QTP is highly sensitive to climate change and the plateau ecosystems are vulnerable to climate warming because about 55% of plateau surface is underlain by seasonally frozen ground (SFG), and 40% to 46% by permafrost (Zou et al., 2017; Cao et al., 2019b, 2022). In most locations in the world, permafrost monitoring sites have revealed an increase trend in permafrost temperature (Biskaborn et al., 2019). On the QTP, the temperature of permafrost at 6 m in depth increased by 0.43 °C from 1996 to 2006 (Wu and Zhang, 2008) and at an average rate of 0.02 °C/year from 2006 to 2010 (Wu et al., 2012). At about half of the permafrost monitoring sites on the QTP, permafrost is warm (> −1 °C), while for about two-thirds of sites, permafrost is warmer than −1.5 °C (Wu et al., 2010).

Earth system models predicted that permafrost thicker than 10 m covers 36% of the QTP and permafrost will continue to thin at annual rates of up to 21 cm/year under various climate change scenarios (Zhao et al., 2022). Several models for permafrost distribution have been built and applied for the QTP but with significant across-model differences in model accuracies as compared to *in-situ* observations (e.g., Yin et al., 2016; Zhao et al., 2017; Zou et al., 2017; Qin et al., 2020; Cao et al., 2022). However, the borehole networks for permafrost monitoring are sparse on the QTP, which are concentrated solely along the main infrastructure lines, and; thus, the monitoring results cannot properly validate the permafrost models at diverse spatiotemporal scales. Therefore, the observational networks of ground surface temperature (GST) could represent a solution to better evaluate these models in rugged terrains and at a finer spatial resolution.

GST is essential for understanding the surface energy dynamics and flows, water cycles, ecology, ecohydrology, and climate change impacts in the Earth Critical Zone, especially in cold regions. GST directly impacts the chemical and biological processes of carbon and nitrogen in the soil and establishes the quality of soil resources (Li et al., 2022). GST has been largely used to understand the thermal regimes in the periglacial environment, the hydrothermal and biogeochemical dynamics of the active layer, and to delineate the distribution of SFG and permafrost (Rödder and Kneisel, 2012; Vieira et al., 2017; Luo et al., 2019; Cao et al., 2019a; Wani et al., 2020; Serban et al., 2021; Jiao et al., 2023). GST influences surface water cycles and vegetation growth and can be used as the upper boundary condition for simulating the thermal state of permafrost and active layer thickness (Wani et al., 2020; Jiao et al., 2023). Thus, GST is monitored through the Global Terrestrial Networks for Permafrost (Biskaborn et al., 2015), and has been recently incorporated into the global database of near-surface temperature (SoilTemp) that covers all types of environments (Lembrechts et al., 2020). Although GST started to be manually measured since the 1950s through the network of the China Meteorological Administration, these earlier measurements were inconsistent with the recent automatic measurements. Furthermore, the manual protocol of historical measurements was highly biased by the presence of snow cover (Cao et al., 2023; Cui et al., 2020). GST is usually measured at approximately 5 cm in the ground but in literature, the GST depth was varying from 2 to 10 cm (Onaca et al., 2015; Ferreira et al., 2017; Oliva et al., 2017;

Grünberg et al., 2020). The GST depth varies according to the landcover type being closer to the ground surface in fine soils and up to 10 cm in pebbles (e.g., rock glaciers, blockfields, protalus ramparts, and scree slopes).

A high spatial variability of the mean annual GST (MAGST) has been reported in alpine regions due to the large heterogeneity of surface covers, topography, and snow cover conditions. However, few studies compared the variability of MAGST at a small scale, such as 0.5 km$^2$ in the Scandinavian Mountains and Arctic Canada (Gisnås et al., 2014; Grünberg et al., 2020), and distances < 50 m in the Swiss Alps (Gubler et al., 2011; Rödder and Kneisel, 2012) or up to 100 m in Southern Norway (Isaksen et al., 2011). Recently, variations in MAGST have started to be also assessed at small scales on the northeastern QTP, such as 3.5 km$^2$ (Luo et al., 2020) and at distances from 2 to 16 m and areas of 2 to 50 km$^2$ (Șerban et al., 2023). Unfortunately, GSTs are still scantly monitored in mountains, while studies have revealed their importance in the cryosphere, ecosystems monitoring, and hazard mitigation (Messenzehl and Dikau, 2017; Hagedorn et al., 2019; Wani et al., 2020; Li et al., 2022). Nevertheless, we lack systematic monitoring of GSTs and an understanding of their fine scale variability in time and space. This paper presents a unique dataset of GST measurements at various spatial scales and under different landcover types in the Headwater Area of the Yellow River (HAYR). The objectives of this paper are: (1) Present the GST monitoring design from fine-scale (2 to 16 m distance) to local and landscape scales (2 and 50 km$^2$) and across an 800-m elevational transect; (2) Evaluate the data quality; (3) Identify the fine-scale variability of GST under similar and different landcover conditions, and; (4) Assess the possibility to identify permafrost occurrence based on GST. This work focuses on presenting the data and mainly the differences of the daily GST at the fine scale. The variability of the MAGST at other scales and their environmental controls have been assessed in detail by Șerban et al. (2023).

## 2 Materials and methods

### 2.1 Study area

The HAYR, situated on the northeastern QTP, is part of the Sanjiangyuan (Source Area of the Three Rivers – SATRs: Yangtze, Yellow, and Lancang-Mekong rivers) National Park and one of the key water towers in China (Fig. 1). The HAYR is an important water conservation zone and a representative area of discontinuous alpine permafrost on northeastern QTP. Numerous thermokarst lakes and ponds occur in this area. For example, on the Chalaping plateau in an area of 150 km$^2$, we mapped 966 water bodies (Șerban et al., 2020). Therefore, the HAYR is a sensitive area to permafrost and ecological degradation under a warming and drying climate (Jin et al., 2009, 2022). Under different climate change scenarios, model predictions show a persistent reduction of permafrost extent in the HAYR by up to 62% by 2100 (Sheng et al., 2020). In the HAYR, the climate is cold and dry, with a mean annual air temperature (MAAT) below –4 °C and annual precipitation at 300-450 mm. Solar radiation is strong throughout a year with an annual evaporation potential at 1000–1500 mm (Jin et al., 2009). The monitoring network of GST is situated in the south-central part of the HAYR at the northern flank of the Bayan Har Mountains and covers an elevational range from 4300 to 5085 m a. s. l. The study area is on a high plateau with smoothed interfluves and peaks and the illumination conditions do not differ substantially. The principal vegetation types in the Bayan

Har Mountains are alpine meadows and alpine swamp meadows. Alpine steppes are present mainly at the feet of the mountains while at higher elevations scree slopes and block streams also occur. However, areas of sparse vegetation and bare ground occur everywhere due to the recently intensified degradation of alpine grasslands (Li et al., 2016; Jin et al., 2022).

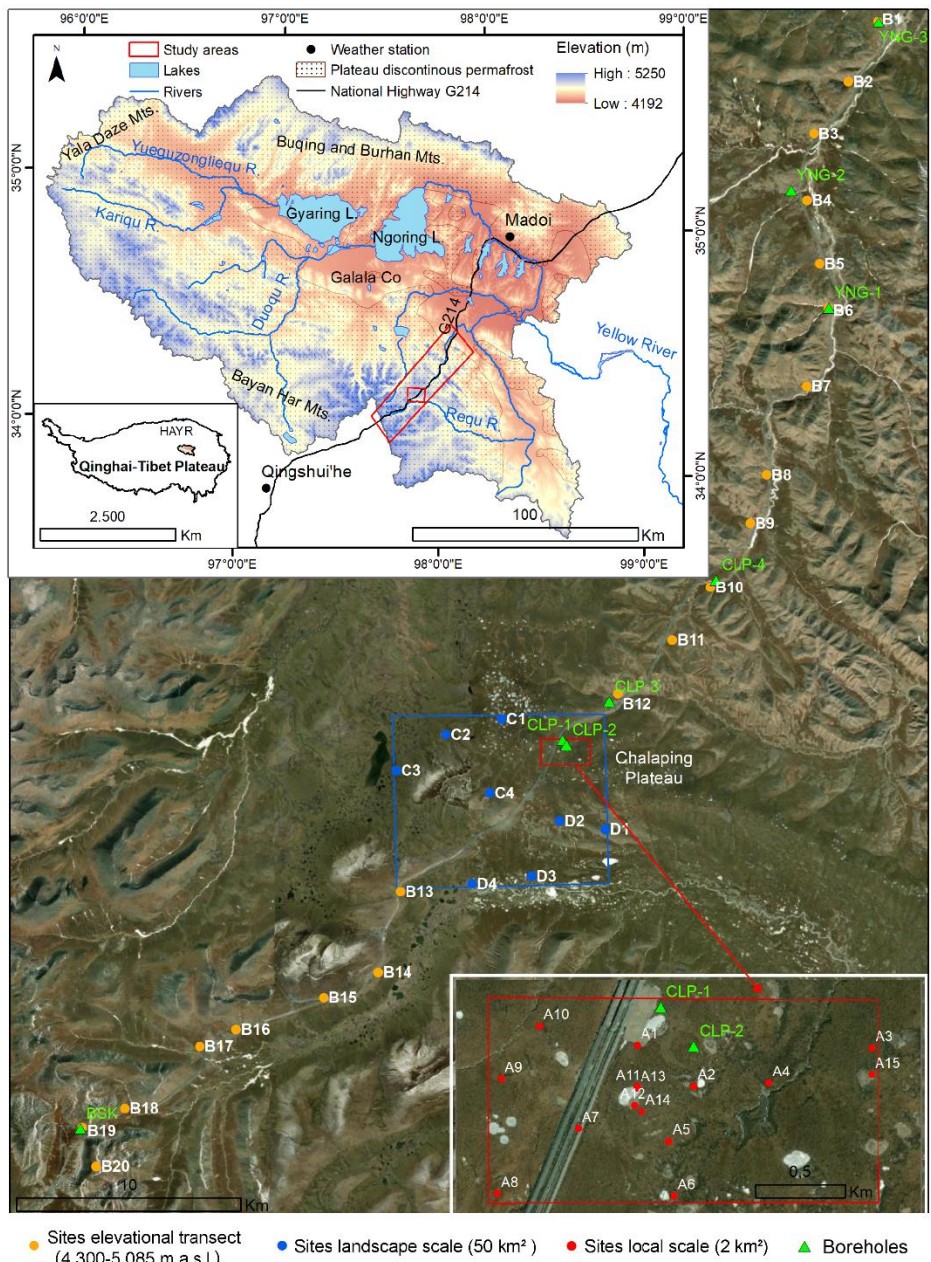

**Figure 1. Location of the monitoring sites of ground surface temperature (GST) in the south-central part of the Headwater Area of Yellow River (HAYR) on northeastern Qinghai-Tibet Plateau (QTP). Permafrost distribution was adopted after Wang et al. (2005). Digital Elevation Model ALOS PALSAR was downloaded from Alaska Satellite Facility (https://www.asf.alaska.edu/) and the background satellite image from ESRI.**

## 2.2 Design of monitoring networks

The observational network of GST in the HAYR was designed and implemented for monitoring the effects of changing landcover and microtopography on the GST regime in a warming permafrost environment. Therefore, GST has been monitored
in different landcover types, such as alpine steppes, meadows, swamp meadows, and bare grounds. In terms of microtopography, GST is monitored mostly on flat terrains but also in disturbed grounds by highway construction, thermokarst depressions, between thermokarst ponds, earth hummocks, and near gullies. In our study, GST was monitored along an 800-m elevational transect, at local and landscape scales (2 and 50 km$^2$), and at a fine scale on distances up to 16 m.

### 2.2.1 Elevational transect (800 m elevation difference)

The elevational transect is a relatively linear transect at the northern flank of the Bayan Har Mountains along the National Highway G214 (Fig. 1). The transect consists of 20 sites (named B sites), covering a distance of 51 km with neighboring intra-site intervals at 2–3 km. The first site is located near Ye'niugou Village at 4331 m a. s. l., and the highest site is on the summit at 5085 m a. s. l. Therefore, the elevational transect covers an elevational difference of approximately 800 m. Sites are placed in the proximity of both sides of the highway in different landcover types. The boreholes for permafrost monitoring along
Highway G214 revealed SFG at the Ye'niugou Village and up to 4400 m a. s. l. Permafrost started to occur at elevations from above 4440 m a. s. l. and up to the Bayan Har Mountains Pass (4833 m a. s. l.) during the observational period of 2010-2017 (Luo et al., 2018b).

### 2.2.2 Local scale (2 km$^2$)

In the middle of the elevational transect on the Chalaping plateau, 13 sites (named A) are placed within an area of 2 km$^2$ for
monitoring the GST at the local scale (Fig. 1). At Chalaping, there is one of the coldest sites of permafrost monitoring in the HAYR. For the period 2010–2017, the measured permafrost thickness was 74 m and the mean annual ground temperatures at 10 and 20 m depths were –1.8 and –1.6 °C , respectively (Luo et al., 2018b). Within the 2 km$^2$, sites are placed in homogenous topographical conditions over a flat peat plateau with an elevational difference of only 18 m (from 4714 to 4732 m a. s. l.). Differentiation is caused by micro-topography and landcover variety, while the linear distance between sites ranges from 70
to 465 m, with an average of 275 m.

### 2.2.3 Landscape scale (50 km$^2$)

The landscape scale is represented by a wider area covering around 50 km$^2$ on the Chalaping plateau (Fig. 1). In this area, the GST is monitored at eight sites (named C and D sites) located in different landcover types similar to the sites from the local scale. The difference is that these sites are placed in pristine natural conditions without any visible anthropic disturbances.
These sites are located over a larger elevational gradient of 181 m (from 4597 to 4778 m a. s. l.) and at an average linear distance of 1.15 km.

#### 2.2.4 Fine scale (2 to 16 m distance)

The GST is measured at each site by two sensors placed in two plots (named A and B) at a variable intra-plot distance ranging from 2 to 16 m (Fig. 2a). This was done due to backup reasons and to identify the variations in GST at a fine scale. For 26 sites, the landcover in the two plots is similar, while that for the others is different (e.g., bare ground *versus* meadow, swamp meadow, or steppe). Bare ground was classified when the vegetation cover in the plots was visually estimated below 10 %. Mainly, the bare ground was represented by fine soils while at a few sites, it was covered by coarse gravels (e.g., B20 and D1).

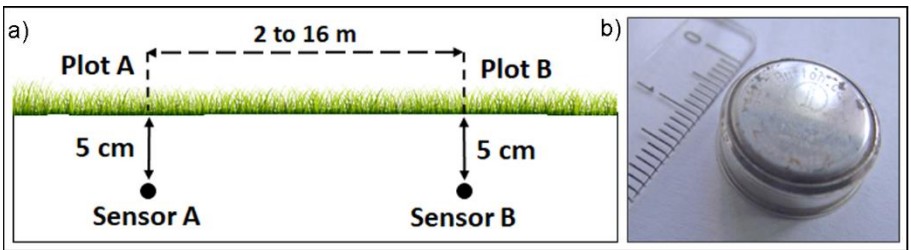

**Figure 2. The schematic draw with the monitoring of GST at the fine-scale: the position of the two plots at distances from 2 to 16 m (a). Photo of the iButton temperature logger DS1922L (b).**

#### 2.3 Instrumentation and measurements

GST was measured at 41 sites and 83 plots using iButton temperature loggers DS1922L. The data loggers have a storage capacity of 8 kB and measure temperatures from –40 to +80 ℃ at a resolution of 0.0625 ℃ and an accuracy of 0.5 ℃. Previous studies performed comparisons and showed that these sensors measure the temperature around zero degrees at an accuracy between –0.12 and 0.30 ℃ (Gubler et al., 2011; Grünberg et al., 2020). These sensors are small (a diameter of 17 mm and a thickness of 6 mm) and easy to deploy in harsh environments (Fig. 2b). Thus, iButtons have been largely used in the Arctic (Way and Lewkowicz, 2018; Goncharova et al., 2019; Grünberg et al., 2020), Antarctica (Ferreira et al., 2017; Oliva et al., 2017; Hrbáček et al., 2020; Lim et al., 2022), and alpine environments (Gubler et al., 2011; Gisnås et al., 2014; Bosson et al., 2015; Colombo et al., 2020).

For deploying these sensors, we followed the standard procedures described in previously mentioned studies. All sensors were sealed in pouches to enhance their waterproofing and installed at a depth of approximately 5 cm (Gubler et al., 2011). This is deep enough to avoid direct solar radiation and close enough to the surface to record the thermal effect of surface covers. The measurement period was one hydrological year from 1 August 2019 to 31 July 2020 and the temperature was recorded once every three hours. The datasets are openly available in the National Tibetan Plateau/Third Pole Environment Data Center (https://dx.doi.org/10.11888/Cryos.tpdc.272945, Șerban and Jin, 2022). The position of each plot was landmarked with wooden and steel sticks and recorded with a handheld GPS with an approximately 3 m accuracy. Photographs were taken at each site and plot (Fig. 3), and a metal detector was used for easier retrieval that assured an 100% success rate.

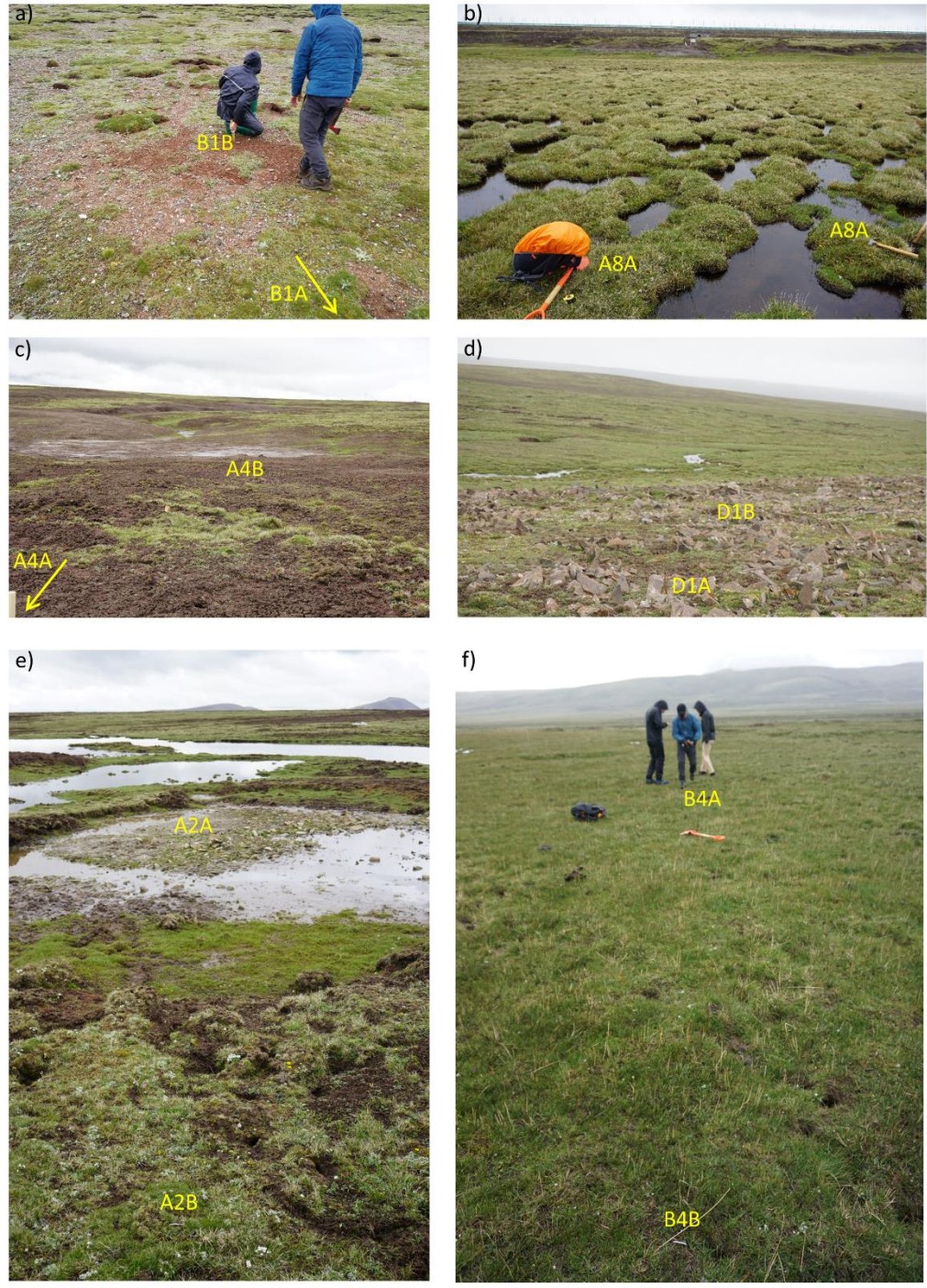

**Figure 3. Photographs presenting the monitoring plots of GST in different landcover types: alpine steppe and bare ground – B1 (a); earth hummocks in alpine swamp meadow – A8 (b); fine bare ground – A4 (c); coarse bare ground – D1 (d); fine bare ground in the depression of a drained thermokarst pound and in the nearby alpine meadow – A2 (e); alpine meadow – B4 (f)."**

From 11 plots, soil samples were collected (Table 1) for grain size and water content analysis. Samples were weighed before
and after being dried at 105 °C for 16 hours to determine the water content. The coarse texture represented by gravel (> 2 mm)
was quantified by sieving, while the fine texture (< 2 mm) representing sand, silt, and clay was measured with a Malvern
Mastersizer-2000 laser diffraction particle size analyzer.

## 2.4 GST data pre-processing and analysis

The raw GST data were processed by adapting the R scripts developed by Way and Lewkowicz (2018) for automated
manipulation of data from the iButtons loggers. With these scripts were calculated the daily, monthly, and annual values of
GST, and the freezing and thawing degree days (FDD and TDD). The FDD represents the annual sum of negative mean daily
GSTs, presented as cumulated negative degree-days (°C·day). The FDD is a measure of the ground cooling during winter, that
facilitates comparisons between cold seasons and can supply a quantified input for modelling. The opposite is the TDD
calculated as the annual sum of positive mean daily GSTs, and displayed as cumulated positive degree-days (°C·day)
(Guglielmin et al., 2008; Oliva et al., 2017; Serban et al., 2021). Based on FDD and TDD, the frost number (FN) is calculated
as follows (Nelson and Outcalt, 1987):

$$FN \; = \; \frac{\sqrt{FDD}}{\sqrt{FDD}+\sqrt{TDD}} \tag{1}$$

An FN greater than 0.5 denotes more FDD than TDD and indicates a dominating freezing regime that favors permafrost
thermal stability. On the opposite, an FN smaller than 0.5 indicates a prevailing thawing that favors the thickening of the active
layer and the ensued permafrost degradation (Hrbáček et al., 2020). However, FN is not an absolute confirmation of permafrost
presence, which depends on the local ground conditions, especially the surface and thermal offsets to the top of permafrost
(Heggem et al., 2006; Rödder and Kneisel, 2012). The surface offset is mainly driven by snow cover and solar radiation and
controlled by topography and vegetation. Thermal offset is mainly controlled by heat transfer and influenced by different soil
thermal conductivities in the frozen and thawed states determined by soil properties, such as soil texture and soil moisture, and
organic contents  (Smith and Riseborough, 2002; Wani et al., 2020).

Several statistical parameters were computed to compare the GST from the two plots of each site: the Pearson correlation
coefficient ($r$), the absolute differences in daily GST and MAGST, and the coefficient of determination ($R^2$) from linear
regression models. The linear models were assessed using the root mean square error (RMSE), mean absolute error (MAE),
and the Akaike information criterion (AIC). The AIC is a statistical test used to assess how well the model fits the data (Akaike,
1974). The Pearson correlation coefficient was set at three levels of significance $p < 0.001$, $p < 0.01$, and $p < 0.05$. The pre-
processing steps, statistical analyses, and graphs were performed using R v4.1.3. Site and plot characteristics and their selected
thermal characteristics are detailed in Table 1.

**Table 1. Sites characteristics in the south-central HAYR. Elev. = Elevation, Dist. = Intra-plot distance, MAGST = mean annual ground surface temperature, FDD or TDD = freezing or thawing degree days, BG = bare ground, AM = alpine meadow, ASM = alpine swamp meadow, AS = alpine steppe, * = soil sample, and NA = Not available.**

| Scale | Site | Elev. (m) | Dist. (m) | Landcover Plot A | Landcover Plot B | MAGST Plot A | MAGST Plot B | FDD Plot A | FDD Plot B | TDD Plot A | TDD Plot B | FN Plot A | FN Plot B |
|---|---|---|---|---|---|---|---|---|---|---|---|---|---|
| LOCAL | A1 | 4720 | 16 | ASM | BG | -0.75 | -1.39 | -1105.83 | -1448.21 | 831.01 | 941.03 | 0.54 | 0.55 |
| | A2 | 4725 | 8 | BG* | AM | -2.18 | NA | -1604.59 | NA | 806.93 | NA | 0.59 | NA |
| | A3 | 4724 | 10 | ASM | ASM | -3.13 | -0.99 | -1952.35 | -1118.35 | 806.56 | 754.36 | 0.61 | 0.55 |
| | A4 | 4715 | 10 | BG | BG | NA | -2.02 | NA | -1396.03 | NA | 656.59 | NA | 0.59 |
| | A5 | 4729 | 10 | BG* | ASM | -1.69 | -0.82 | -1439.58 | -1281.63 | 819.61 | 981.41 | 0.57 | 0.53 |
| | A6 | 4728 | 8 | ASM | BG* | -1.13 | -1.19 | -1296.93 | -1300.41 | 884.08 | 864.54 | 0.55 | 0.55 |
| | A7 | 4721 | 8 | AM | AM | 0.31 | 0.54 | -936.5 | -947.525 | 1049.04 | 1144.25 | 0.49 | 0.48 |
| | A8 | 4732 | 2 | ASM | ASM | -0.95 | -1.33 | -1281.56 | -1238.51 | 932.80 | 752.96 | 0.54 | 0.56 |
| | A9 | 4726 | 8 | BG | ASM | -2.02 | -1.44 | -1434.69 | -1318.13 | 695.53 | 792.26 | 0.59 | 0.56 |
| | A10 | 4719 | 14 | BG* | ASM | -2.49 | -0.59 | -1621.78 | -1283.49 | 710.49 | 1068.23 | 0.60 | 0.52 |
| | A12 | 4728 | 14 | BG* | AM | -1.56 | -0.59 | -1405.35 | -1132.46 | 835.61 | 914.90 | 0.56 | 0.53 |
| | A15 | 4722 | 20 | BG | AM | NA | -0.30 | NA | -1083.23 | NA | 974.38 | NA | 0.51 |
| LANDSCAPE | C1 | 4737 | 10 | ASM | ASM | NA | -0.39 | NA | -1203.89 | NA | 1061.13 | NA | 0.52 |
| | C2 | 4704 | 2 | ASM | ASM | -0.23 | -0.46 | -1060.03 | -1101.91 | 976.80 | 932.24 | 0.51 | 0.52 |
| | C3 | 4668 | 4 | BG* | ASM | -1.36 | 0.17 | -1238.61 | -946.575 | 739.03 | 1008.11 | 0.56 | 0.49 |
| | C4 | 4778 | 14 | ASM | ASM | -2.33 | -1.33 | -1560.01 | -1305.79 | 706.53 | 819.54 | 0.60 | 0.56 |
| | D1 | 4645 | 8 | BG | BG | -2.18 | -1.65 | -1664.2 | -1579.48 | 865.28 | 975.30 | 0.58 | 0.56 |
| | D2 | 4666 | 8 | ASM | ASM | 1.49 | 0.72 | -798.45 | -760.113 | 1344.99 | 1024.71 | 0.44 | 0.46 |
| | D3 | 4598 | 14 | BG | BG | -1.17 | -1.19 | -1301.36 | -1187.86 | 871.99 | 753.45 | 0.55 | 0.56 |
| | D4 | 4598 | 6 | AM | AM | 0.84 | 0.68 | -874.888 | -854.788 | 1183.66 | 1102.10 | 0.46 | 0.47 |
| ELEVATIONAL TRANSECT | B1 | 4331 | 6 | AS | BG | 1.15 | 0.87 | -919.313 | -998.125 | 1338.04 | 1317.29 | 0.45 | 0.47 |
| | B2 | 4376 | 14 | AM* | AM | 1.33 | NA | -762.95 | NA | 1248.78 | NA | 0.44 | NA |
| | B3 | 4333 | 2 | BG* | BG | 1.26 | 1.32 | -976.663 | -923.163 | 1436.96 | 1405.19 | 0.45 | 0.45 |
| | B4 | 4340 | 10 | AM | AM | 2.29 | 2.26 | -536.663 | -493.963 | 1374.31 | 1317.70 | 0.38 | 0.38 |
| | B5 | 4357 | 4 | AM* | AM | 1.66 | 1.13 | -775.825 | -835.35 | 1383.04 | 1248.57 | 0.43 | 0.45 |
| | B6 | 4399 | 2 | ASM | ASM | 1.88 | 1.82 | -616.5 | -651.6 | 1304.34 | 1315.82 | 0.41 | 0.41 |
| | B7 | 4432 | 14 | BG | BG* | 1.14 | 1.45 | -859.138 | -805.613 | 1277.29 | 1335.59 | 0.45 | 0.44 |
| | B8 | 4473 | 10 | BG* | BG | 0.21 | 0.39 | -1023.53 | -1006.1 | 1099.67 | 1150.05 | 0.49 | 0.48 |
| | B9 | 4496 | 6 | AM | AM | 0.51 | 0.26 | -867.163 | -987.175 | 1053.28 | 1083.55 | 0.48 | 0.49 |
| | B10 | 4565 | 2 | ASM | ASM | 1.25 | 1.25 | -602.438 | -673.525 | 1058.16 | 1132.81 | 0.43 | 0.44 |
| | B11 | 4607 | 8 | ASM | BG | 0.06 | -0.96 | -1023.19 | -1306.56 | 1046.45 | 954.99 | 0.50 | 0.54 |
| | B12 | 4643 | 2 | ASM | ASM | -0.42 | -0.17 | -1105.4 | -1131.5 | 953.46 | 1070.13 | 0.52 | 0.51 |
| | B13 | 4642 | 12 | ASM | ASM | 1.66 | 1.47 | -537.5 | -596.525 | 1144.90 | 1118.09 | 0.41 | 0.42 |
| | B14 | 4637 | 10 | AM | AM | 0.50 | 0.77 | -750.925 | -742.45 | 934.69 | 1025.76 | 0.47 | 0.46 |
| | B15 | 4686 | 6 | AM | BG | 1.06 | 1.23 | -824.063 | -837.538 | 1213.43 | 1286.84 | 0.45 | 0.45 |
| | B16 | 4672 | 2 | ASM | ASM | 2.22 | 1.87 | -506.238 | -591.375 | 1320.34 | 1274.85 | 0.38 | 0.41 |
| | B17 | 4665 | 8 | ASM | ASM | 0.83 | 0.32 | -763.213 | -900.638 | 1067.93 | 1016.53 | 0.46 | 0.48 |
| | B19 | 4833 | 14 | AM | BG | NA | -0.75 | NA | -1229.58 | NA | 954.39 | NA | 0.53 |
| | B20 | 5085 | 6 | BG | BG | -4.69 | -4.29 | -2170.91 | -2102.7 | 455.01 | 533.61 | 0.69 | 0.66 |

# 3 Results and discussions

## 3.1 Data quality check

The datasets were quality checked in order to detect and remove measurement errors. Among the measured data from all 83 sensors, those from 11 sensors were excluded in the dataset because they did not have complete time series. Among the 11 malfunctioned sensors, four sensors had become malfunctioned without any recorded measurements, while three sensors stopped recording the measurements seven months after installations. One sensor was found on the ground surface due to frost-jacking and exposed to direct solar radiation. Other three sensors placed on the shore of a thermokarst lake; one year after

installations, they were found to have submersed approximately 10 cm under the lake water. The HAYR has a dynamic landscape, where thermokarst lakes and ponds have revealed both patterns of expanding and shrinking during the period 1986–2015 (Șerban et al., 2021).

Some studies reported problems with sensor malfunction or retrieval. For example, 13% of the iButtons used to monitor the snowpack in Northwestern Canada failed due to water infiltration (Lewkowicz, 2008), revealing the importance of sealing

them in pouches. In the Swiss Alps, 7% of the sensors deployed to monitor the GST reappeared on the surface and were also exposed to direct solar radiation (Gubler et al., 2011). The iButtons were widely used also in Antarctica to monitor the GST and proved reliable in harsh conditions. Lim et al. (2022) reported that among the 131 sites in Antarctica, only at three sites GST was not recorded due to device errors.

In the monitoring networks of GST in the HAYR, a total of 210,816 measurements were recorded, of which 165 were biased

due to sensor malfunctions. From these, the most severe one was found in plot A3A, with a period of 10 days from 1 to 19 September 2019, with 151 measurements blocked at –41 ℃. In addition, there were another 13 erroneous measurements with temperatures of –41, –39.5, and 87 ℃ on 23 and 26 February 2023. The sensor from plot B16B had only one wrong measurement of –7.7 ℃ on 17 October 2019, while the other temperature readings during that period ranged from 0.1 to 2.6 ℃. All these incorrect measurements were replaced with missing values (NA - not available). In addition, there were other

NA values due to data downloaded with 1–1.5 days earlier than 31 July 2020 for sites B1 to B8.

## 3.2 Variability of GST at the fine scale

For the local and landscape scale settings, the daily intra-plot differences in GST were up to 2 ℃ for the sites with similar landcovers in both plots (Figs. 4 and 5). Differences were usually larger during autumn and the end of spring and the beginning of summer, while for other periods were below 1 ℃. For other sites where one plot was in bare ground and the other under

vegetation, the daily intra-plot differences in GST were frequently exceeding 2 ℃. This happened especially at the end of autumn, the beginning of winter, the end of spring, and the beginning of summer. In a time period of several days (from 1 to 15), the differences exceeded even 4 ℃ at sites A6, A5, A1, A10, and C3. The maximum difference of 5.3 ℃ was on 4 and 5 December 2019 at sites A1 and A10, sites with the largest intra-plot distance of 14 m. At site C3, data of longer periods were recorded from autumn to spring with daily differences in GST between 2 and 4 ℃. There were a few days with differences

higher than 4 °C at sites D2 (6 days in May 2020) and C4 (5 days in December 2019). At these two sites, the plots are situated in a swamp meadow at distances of 8 and 14 m, respectively.

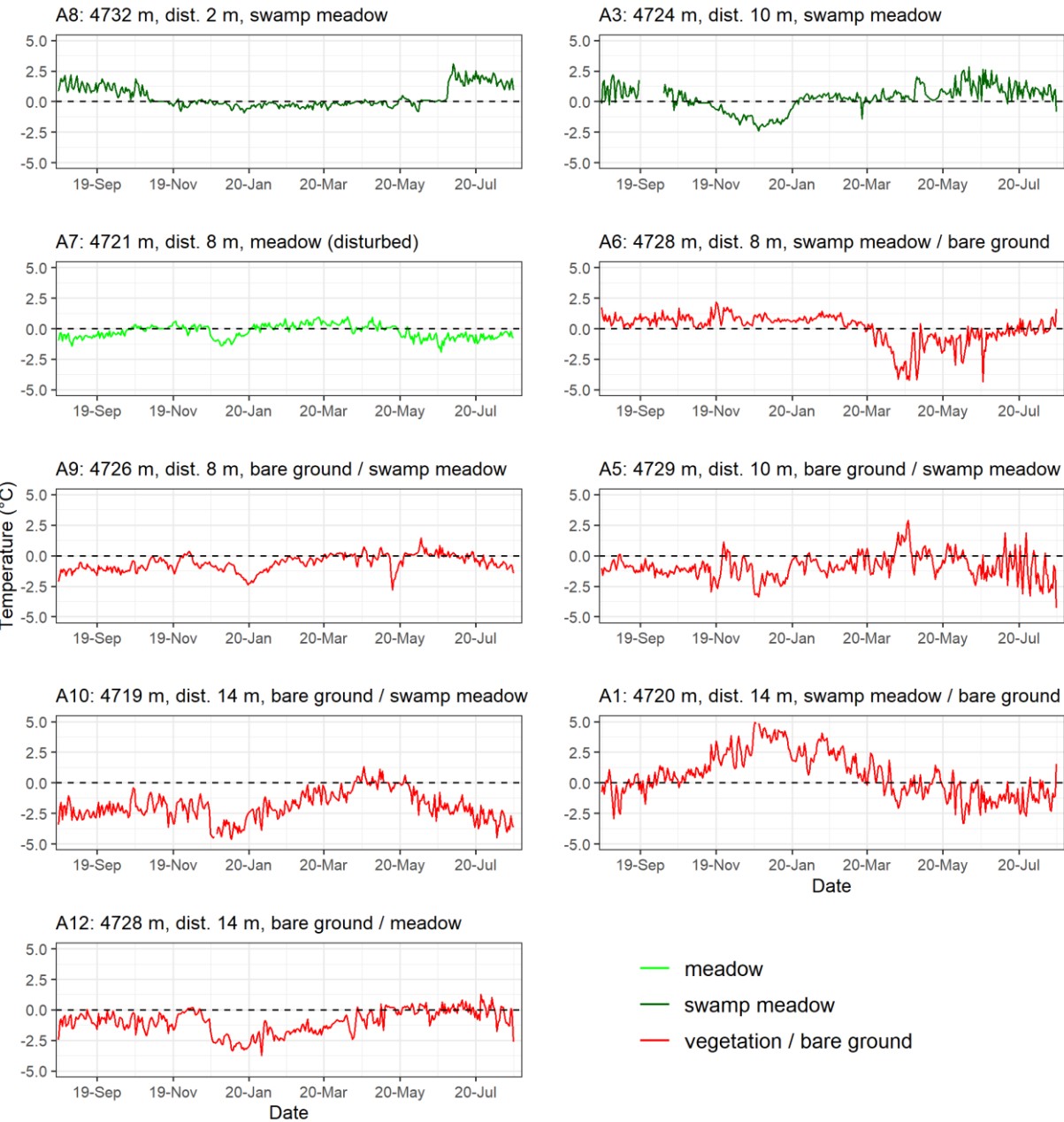

**Figure 4. Intra-plot differences in daily GST for period August 2019 – July 2020 at sites at the local scale (2 km²) on the Chalaping peat plateau in the south-central HAYR.**

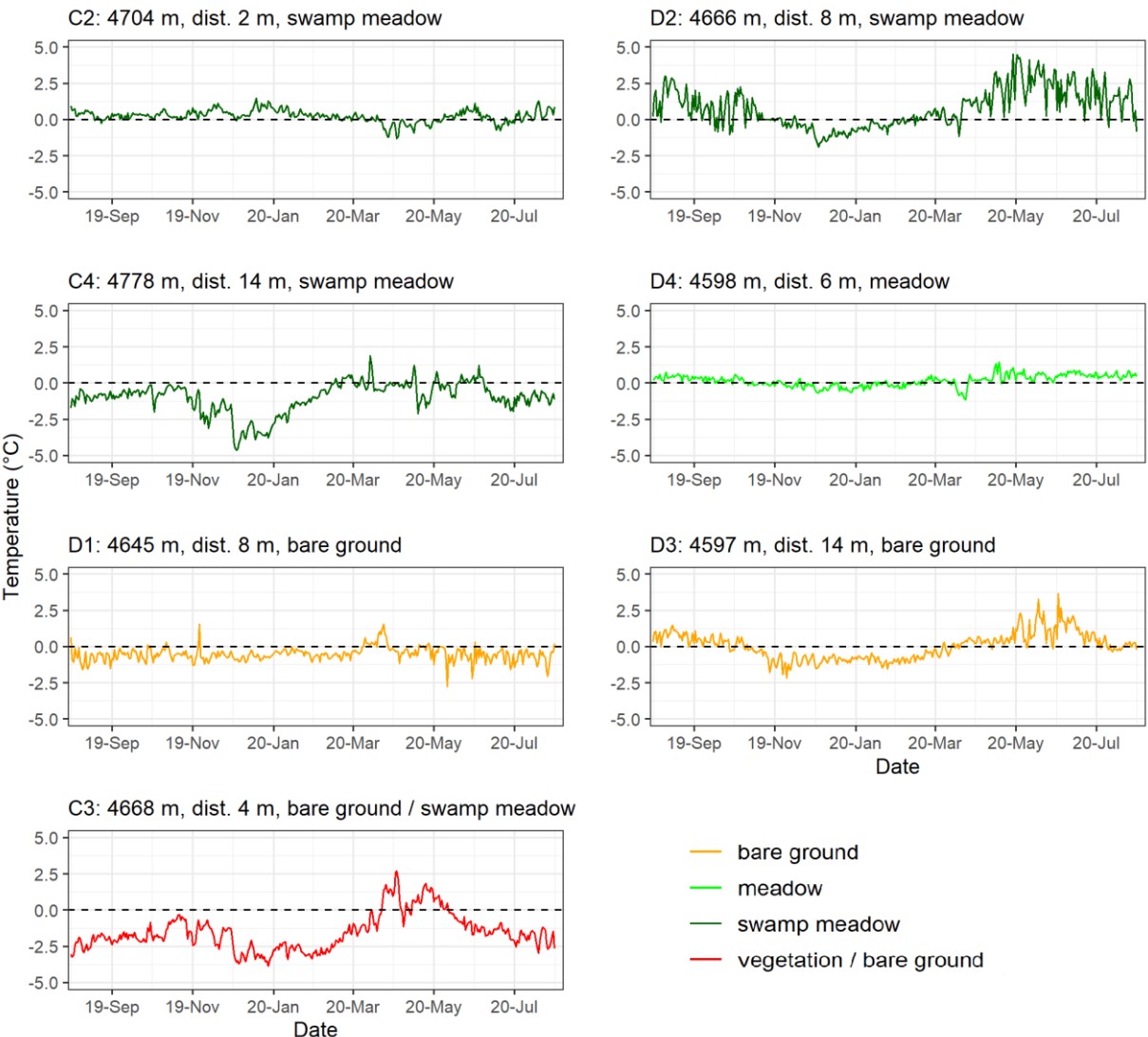


**Figure 5. Intra-plot differences in daily GST for period August 2019 – July 2020 in sites at the landscape scale (50 km$^2$) on the Chalaping peat plateau in the south-central HAYR.**

Most of the sites along the elevational transect presented a similar pattern with an intra-plot difference in daily GST below 2

°C (Figs. 6 and 7). Differences larger than 2.5 °C were observed mainly at the sites at elevations above 4600 m a. s. l., regardless

of landcover types in the plots. At these sites, for 1 to 5 days, the differences exceeded 4 °C, mainly in summer and autumn

(B13, B14, and B16), but also in spring (B13, B15, and B17) and winter (B11). The highest differences of 4.8 °C were recorded

on 4 and 12 October 2019 at sites B13 and B16 with both plots in swamp meadow. While at site B13, it was an intra-plot

distance of 12 m, and; at site B16, only 2 m. The larger intra-plot difference in daily GST at higher elevations may be related

to the temperature inversion observed on the elevational transect. These temperature inversions showed seasonality, being

more visible in spring and especially in winter (Șerban et al., 2023). However, they could also indicate a diurnal variability caused by the strong radiation cooling under dry conditions and the local air circulation. The spatial differences in the reduction of plant species and root biomass could also increase the GST due to the decrease in evapotranspiration (Du et al., 2007).

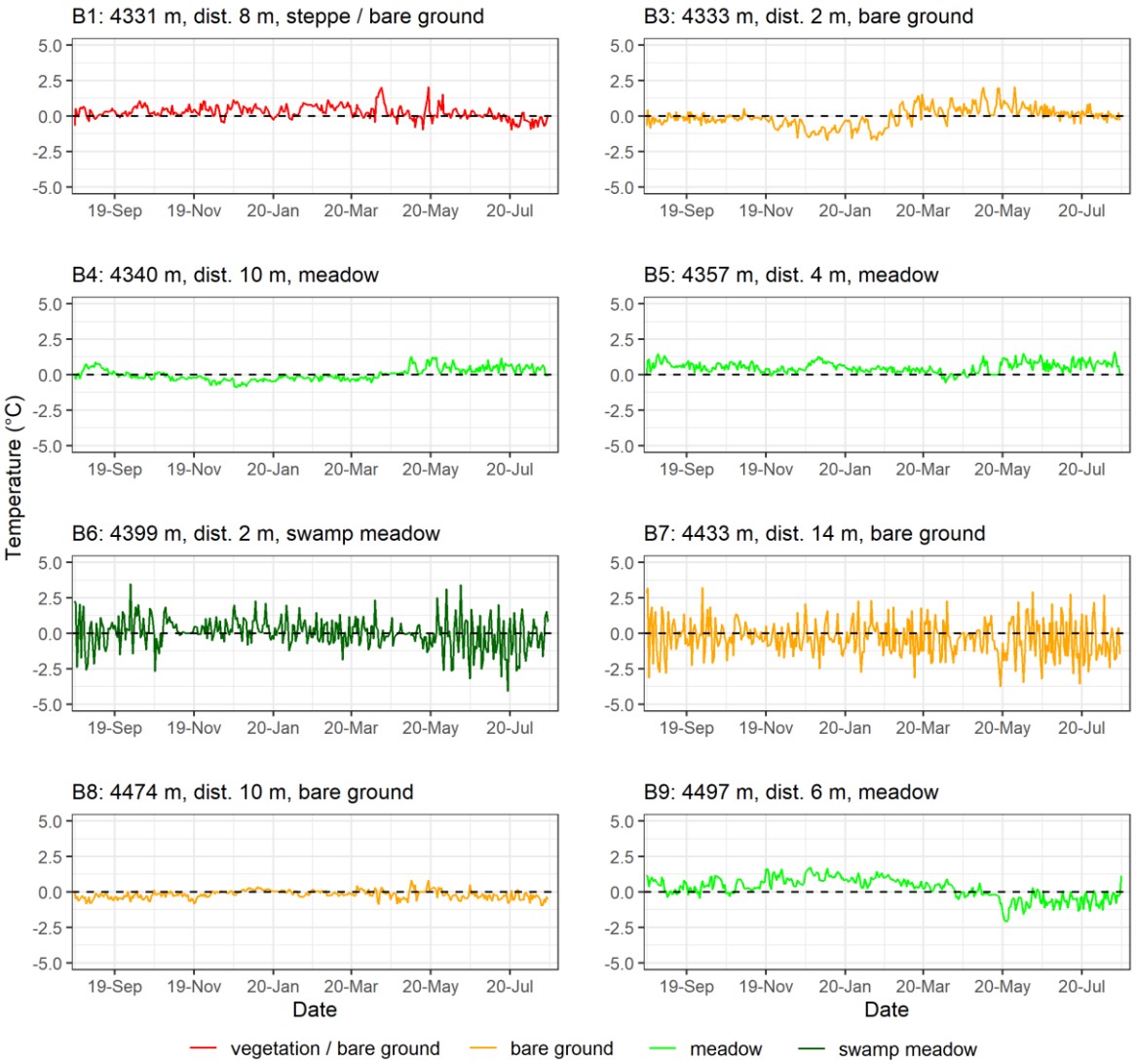


**Figure 6. Intra-plot differences in daily GST for period August 2019 – July 2020 in sites B1 to B9 along the elevational transect in the south-central HAYR.**

The daily GST values revealed a linear relationship between the two compared plots of each site. Correlation coefficients were above 0.962 for all sites and at a significance level of $p < 0.001$. The coefficient of determination ($R^2$) was high, ranging

between 0.926 at site B14 to 0.999 at site B8 (Table 2). Both sites B8 and B14 had an intra-plot distance of 10 m. These sites revealed the lowest and largest RMSE, MAE, and AIC of 0.250, 0.188, and 23.739 at site B8 and 1.500, 1.220, and 1342.126

at site B14. Although the $R^2$ was high, the slightest variation revealed an RMSE of around 1 °C, and; as previously shown, the intra-plot differences in daily GST went as high as 4–5 °C.

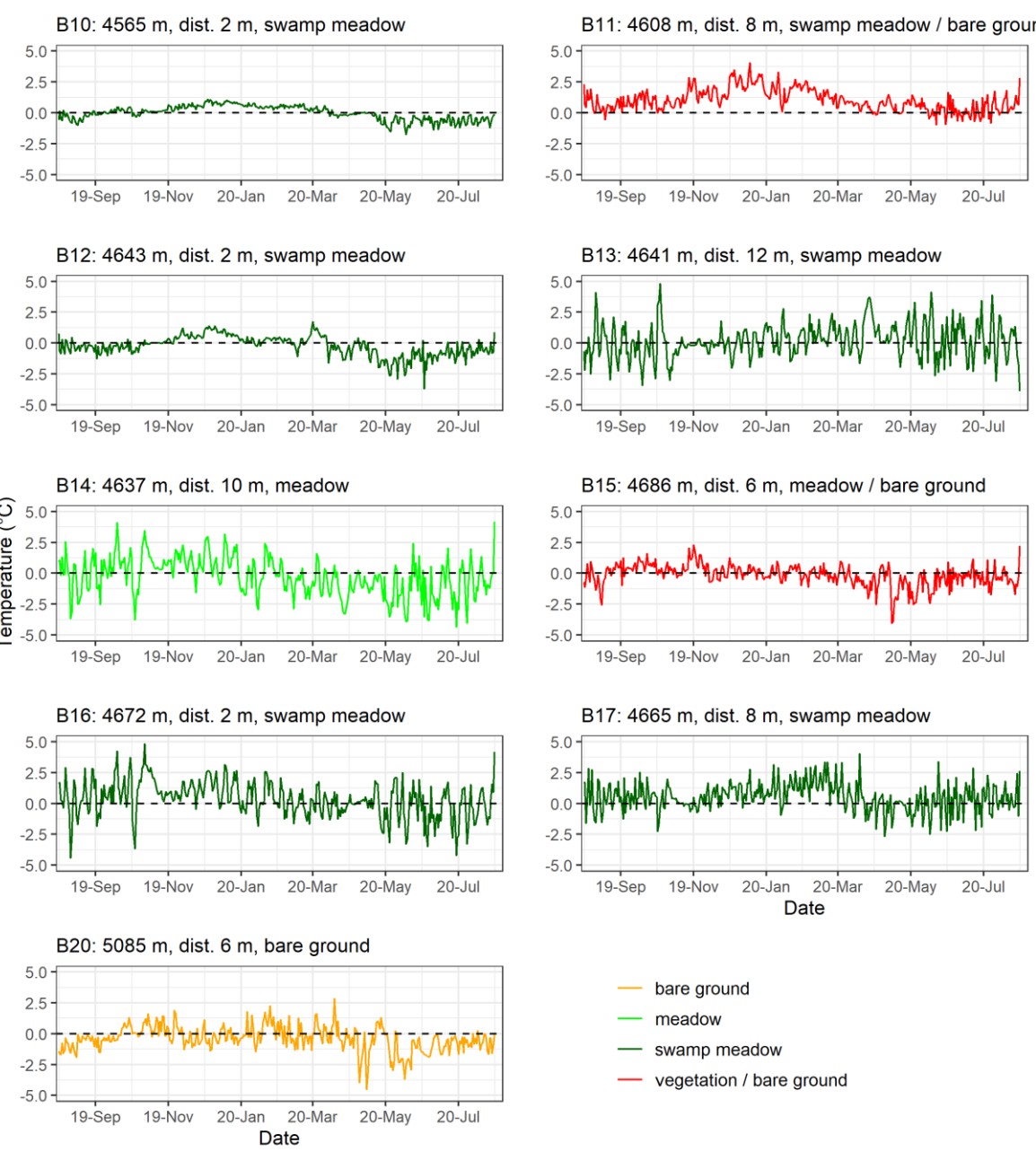

**Figure 7. Intra-plot differences in daily GST for period August 2019 – July 2020 in sites B10 to B20 along the elevational transect in the south-central HAYR.**

**Table 2. Statistical parameters of daily intra-plot comparisons of GST for the period of August 2019 to July 2020 in the south-central HAYR.**

| Scale | Site | $R^2$ | RMSE | MAE | AIC | r | Scale | Site | $R^2$ | RMSE | MAE | AIC | r |
|---|---|---|---|---|---|---|---|---|---|---|---|---|---|
| LOCAL | A1 | 0.968 | 1.114 | 0.917 | 1123.525 | 0.984 | ELEVATIONAL TRANSECT | B1 | 0.997 | 0.400 | 0.298 | 368.220 | 0.998 |
| | A3 | 0.979 | 0.934 | 0.741 | 943.112 | 0.989 | | B3 | 0.993 | 0.640 | 0.495 | 719.838 | 0.996 |
| | A5 | 0.980 | 0.993 | 0.749 | 1039.384 | 0.990 | | B4 | 0.997 | 0.340 | 0.264 | 255.483 | 0.998 |
| | A6 | 0.971 | 1.157 | 0.838 | 1151.100 | 0.985 | | B5 | 0.997 | 0.340 | 0.273 | 260.373 | 0.999 |
| | A7 | 0.995 | 0.437 | 0.333 | 438.255 | 0.998 | | B6 | 0.971 | 1.040 | 0.778 | 1071.205 | 0.985 |
| | A8 | 0.995 | 0.494 | 0.373 | 528.223 | 0.997 | | B7 | 0.967 | 1.230 | 0.966 | 1193.099 | 0.983 |
| | A9 | 0.991 | 0.645 | 0.520 | 723.413 | 0.995 | | B8 | 0.999 | 0.250 | 0.188 | 23.739 | 0.999 |
| | A10 | 0.974 | 1.148 | 0.880 | 1145.850 | 0.986 | | B9 | 0.991 | 0.590 | 0.449 | 659.186 | 0.995 |
| | A12 | 0.988 | 0.790 | 0.661 | 872.185 | 0.994 | | B10 | 0.995 | 0.370 | 0.289 | 322.477 | 0.998 |
| | | | | | | | | B11 | 0.989 | 0.680 | 0.549 | 763.880 | 0.995 |
| LANDSCAPE | C2 | 0.996 | 0.410 | 0.305 | 396.916 | 0.998 | | B12 | 0.989 | 0.690 | 0.522 | 773.471 | 0.994 |
| | C3 | 0.960 | 1.250 | 0.950 | 1207.409 | 0.978 | | B13 | 0.936 | 1.340 | 1.048 | 1259.732 | 0.967 |
| | C4 | 0.975 | 1.070 | 0.801 | 1096.175 | 0.986 | | B14 | 0.926 | 1.500 | 1.220 | 1342.126 | 0.962 |
| | D1 | 0.996 | 0.480 | 0.352 | 504.651 | 0.998 | | B15 | 0.983 | 0.840 | 0.633 | 915.171 | 0.992 |
| | D2 | 0.971 | 1.120 | 0.880 | 1130.655 | 0.986 | | B16 | 0.940 | 1.400 | 1.099 | 1288.713 | 0.970 |
| | D3 | 0.989 | 0.710 | 0.515 | 796.936 | 0.994 | | B17 | 0.965 | 1.080 | 0.830 | 1102.084 | 0.982 |
| | D4 | 0.998 | 0.290 | 0.220 | 136.351 | 0.999 | | B20 | 0.985 | 0.910 | 0.707 | 977.181 | 0.992 |

Larger differences in GST are particularly pronounced in winter when the bare ground cools faster than the vegetated areas (Fig. 8). Three cases were identified: (i) Situations when the bare ground cooled faster in winter, while alpine meadows and alpine swamp meadows warmed faster in summer (e.g., C3, A5, and A10; Fig. 8a); (ii) Bare ground cooled faster in winter and warmed faster in summer than vegetated areas (e.g., A1; Fig. 8b), and; (iii) Bare ground cooled faster in winter, with similar warming like vegetated areas in summer (e.g., A12, A9, and B11; Fig 8c). Almost always, the bare ground was colder in the winter from the end of October until the next April. When comparing the bare ground to alpine steppe, the evolution of GST was similar between the two plots (e.g., B1; Fig. 8d). As previously shown, the intra-plot differences in daily GST shot up as high as 4 °C also when compared with the same landcover in both plots. For example, there were sites with alpine swamp meadows in both plots and in one case the differences were during the warm season (e.g., D2; Fig. 8e) while for the other, during winter (e.g., C4; Fig. 8f). At most of these sites with obvious differences, the intra-plot distance was 14 m. Exceptions were at sites D2, B1, and C3 with intra-site distances between adjacent plots of 8, 6, and 4 m respectively. At site C3 in the valley among thermokarst ponds, the large variability of GST was exemplified across short distances controlled by the landcover types. This site presented one of the largest ranges of the MAGST between –1.36 °C in the bare ground plot and 0.17 °C in the swamp meadow plot.

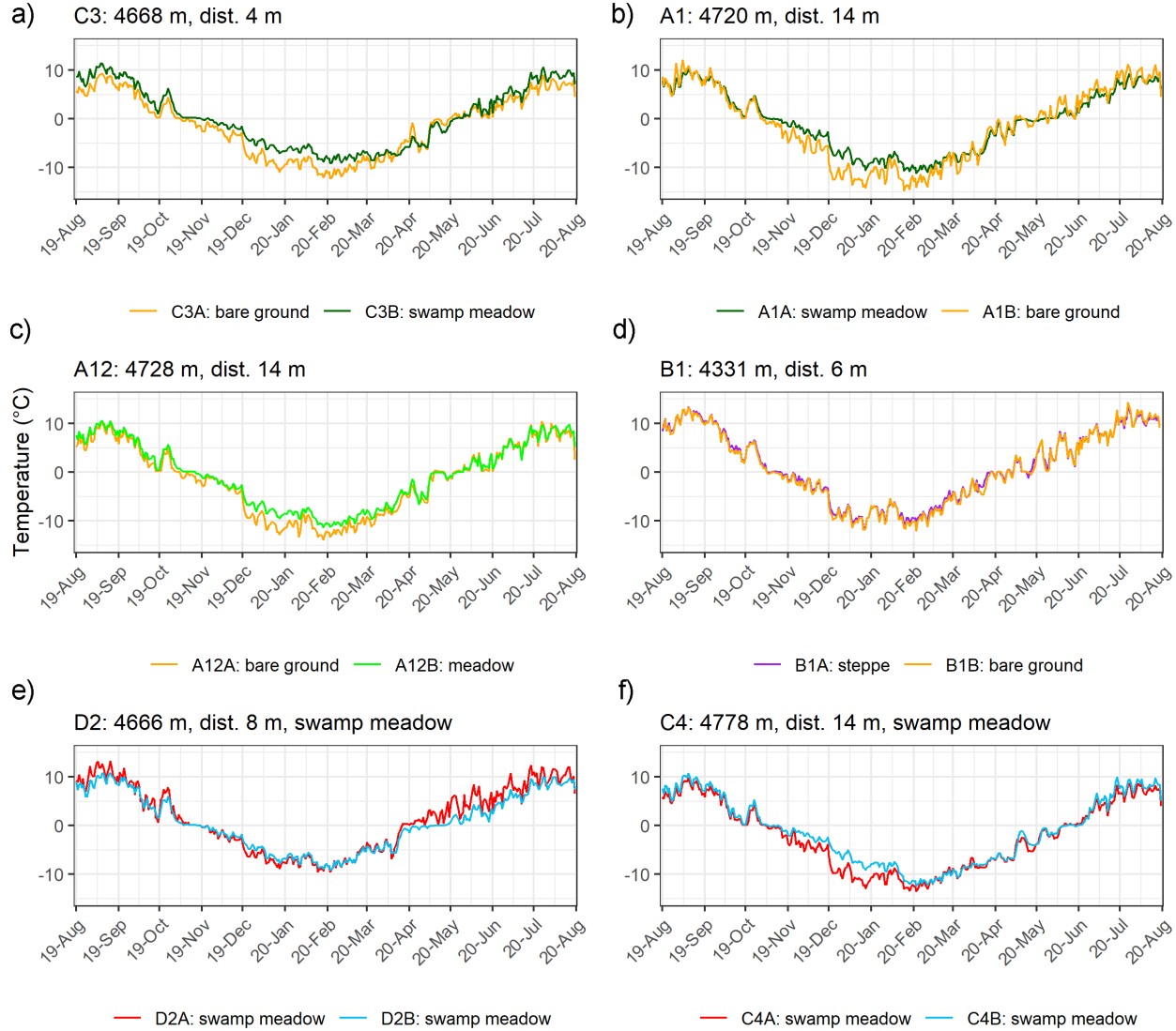

**Figure 8. Evolution of the daily GST for period August 2019 – July 2020 at sites with different landcover between the paired plots: bare ground and alpine swamp meadow (a and b), bare ground and alpine meadow (c), bare ground and alpine steppe (d), and sites with alpine swamp meadow in both plots (e and f) in the south-central HAYR.**

The intra-plot variability of MAGST was mainly below 0.5 °C, especially for sites with the same landcover in both plots (Fig. 9a). Variations between 1 and 2 °C were observed at the sites where the bare ground was compared to vegetated plots (A10, C3, and B11). This variability was observed mainly at all analyzed distances between plots and according to the analysis of variance and linear regression, it was insignificantly influenced by distance (Șerban et al., 2023). The intra-plot variability in the minimum GST was slightly higher, exceeding 2 and 3 °C at sites A1, A10, A12, B11, and C3 (Fig. 9b). At all these sites, the bare ground was compared to meadows or swamp meadows, at intra-plot distances of 14 and 16 m, except at B11 and C3 with those of 8 and 4 m, respectively. The intra-plot variability in maximum GST was above 2 °C only at sites A10, C3, and

D2 (Fig. 9c). Site D2 was the only one with such a large variability in the maximum GST values of –2.44 °C and with swamp meadow in both plots. At this site, the MAGST variability was high (–0.8 °C) at an intra-plot distance of 8 m.

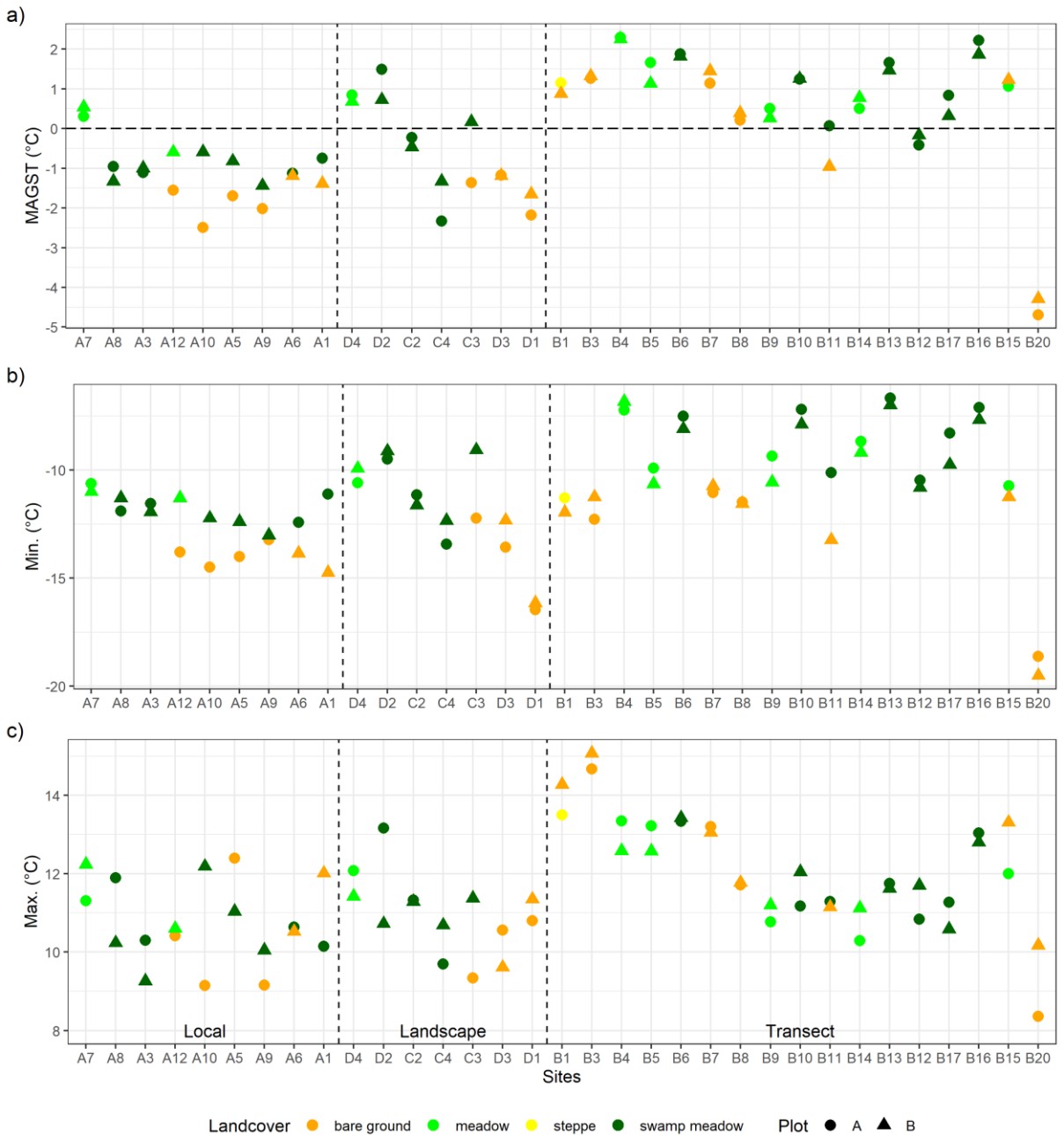


**Figure 9. Mean annual ground surface temperature (MAGST) (a), the minimum GST (b), and the maximum GST (c) for the period August 2019 to July 2020 in both plots A and B of each site in south-central HAYR. Notes: Local scale (2 km²), landscape scale (50 km²), and elevational transect (800 m difference).**

To better understand the variability of MAGST under the influence of landcover, even under the same landcover type, the MAGST was compared with the soil texture and soil water content (Fig. 10, Table 3). All the samples from the local and landscape scales collected from bare ground and with a fine texture of above 75% (Fig. 10a) revealed a low MAGST ranging between –2.2 and –1.2 °C (Fig. 10b). The samples from the lower part of the elevational transect (up to 4432 m) revealed a MAGST between 1.3 and 1.7 °C, regardless of landcover type (meadow or bare ground) or texture. Three of them had a fine

texture between 70 and 98%, except plot B3A with the lowest fine texture of 59%. Only plot B8A from bare ground with a fine texture of 73% had the lowest MAGST of 0.2 °C among the elevational sites sampled for soil texture. However, the elevation of plot B8A is still relatively low, only 4473 m a. s. l.

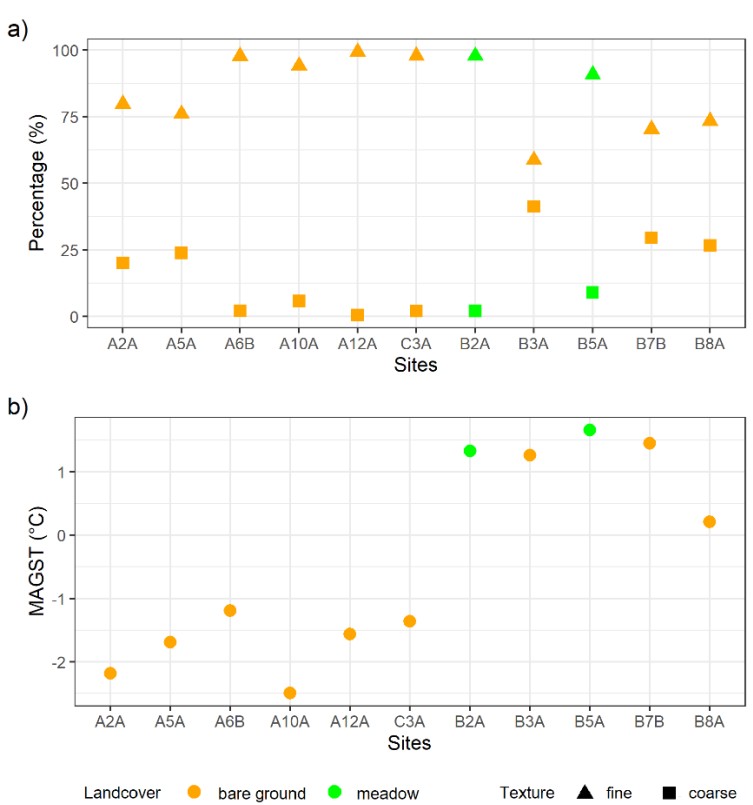


**Figure 10. Textural classification of the soil samples from selected plots (a) and their corresponding mean annual ground surface temperature (MAGST) for the period August 2019 to July 2020 in south-central HAYR (b). Notes: Coarse soil texture is represented by gravel while fine soil texture by sand, silt, and clay, with the threshold between them set at 2 mm in diameter.**


**Table 3. Grain size distribution and water content of the soil samples from selected plots expressed in percentage (%).**

| Site/Plot | Soil texture | | Fine texture details | | | Water content |
|---|---|---|---|---|---|---|
| | Coarse (>2 mm) | Fine (<2 mm) | Sand (>63 μm) | Silt (2 - 63 μm) | Clay (<2 μm) | |
| A2A | 20.14 | 79.86 | 35.93 | 56.65 | 7.42 | 8.15 |
| A5A | 23.96 | 76.04 | 50.18 | 44.71 | 5.11 | 12.64 |
| A6B | 2.19 | 97.81 | 73.21 | 25 | 1.79 | 41.94 |
| A10A | 5.81 | 94.19 | 67.43 | 30.91 | 1.66 | 38.60 |
| A12A | 0.61 | 99.39 | 69.83 | 28.67 | 1.5 | 43.82 |
| C3A | 2.05 | 97.95 | 68.39 | 30.35 | 1.26 | 42.07 |
| B2A | 2.12 | 97.88 | 61.42 | 37.14 | 1.44 | 21.41 |
| B3A | 41.28 | 58.72 | 65.91 | 30.28 | 3.81 | 1.28 |
| B5A | 9.10 | 90.90 | 71.27 | 26.87 | 1.86 | 31.53 |
| B7B | 29.60 | 70.40 | 54.57 | 41.92 | 3.51 | 8.34 |
| B8A | 26.63 | 73.37 | 49.48 | 44.83 | 5.69 | 12.63 |

The detailed comparison of these sites revealed higher MAGST in alpine meadows, followed by alpine swamp meadows, and bare ground. The intra-site MAGST variability has been mainly controlled by elevation and landcover types (as is shown in

Figs. 4 and 8 of Șerban et al., 2023), similar to observation from the Swiss Alps (Gubler et al., 2011). Slope angles and aspects do not play a relevant role because the monitoring plots are located mostly in flat areas (Șerban et al., 2023).

Vegetation plays an important role in better insulating the ground while the lower GSTs in the bare ground reveal the strong coupling with air temperature (Gubler et al., 2011; Aalto et al., 2013). The lower GSTs in the swamp meadows than in dry meadows are caused by the evaporative cooling and the possibly higher thermal offset of the peat soils. In the HAYR,

evaporation was three times higher than precipitation from 1986 to 2015 largely influencing the water balance and changes in thermokarst lakes and ponds (Șerban et al., 2021).

A large spatial variability in MAGST has been observed in other regions. For example, in the Swiss Alps, MAGST varied up to 2.5 and 4.3 °C at a distance of 14 and 50 m (Gubler et al., 2011; Rödder and Kneisel, 2012), and up to 6 °C in an area of 0.5 km$^2$ in the Scandinavian Mountains (Gisnås et al., 2014). Isaksen et al. (2011) reported variations in MAGST of 1.5 to 3 °C

over distances of 30–100 m in isolated patches of cold permafrost, marginal permafrost, and seasonal frost from Southern Norway. These variations were due to blocky terrains, steep slopes, and snow cover thickness and duration. On a small and relatively flat area of 0.5 km$^2$ in the Trail Valley Creek, Canada, MAGST varied up to 3.6 °C due to the variability of vegetation types and snow cover conditions (Grünberg et al., 2020). MAGST variations of up to 2.4 °C were previously observed on the Chalaping plateau but over a larger area of 3.5 km$^2$ (Luo et al., 2020). Furthermore, MAGST increased by 0.06 °C/year from

2011 to 2017 (Luo et al., 2018a). MAGST variations of up to 3 °C were also reported from the Qilian Mountains on the northeastern QTP (Cao et al., 2018). An increase of 0.49–0.43 °C/decade was reported for the entire QTP for soil temperatures at depths of 0–20 cm for the period 1981–2020. These rates were observed from 141 stations and based on the ERA5 reanalysis data (Li et al., 2022). Based on reanalysis products and *in-situ* data, a ground warming rate of 0.0994 °C/year at 10 cm in depth

has been reported along the Qinghai-Tibet Engineering Corridor on interior QTP for the period 2004–2018. In addition, the
ground warming rate was higher in the areas of SFG than in permafrost regions (Jiao et al., 2023).

### 3.3 Variability of FDD, TDD, and FN at the fine scale

While most of the sites presented similar FDD values in both plots, there were few sites with large differences in FDD between plots (Fig. 11a). Surprisingly, at site A3, both plots were in swamp meadow and the intra-plot difference in FDD was around –800 °C·day. A lower difference of around –250 °C·day was also revealed at site C4 with both plots in swamp meadow. Sites A3 and C4 had intra-plot distances of 10 and 14 m, respectively (Table 1). Intra-plot differences in FDD up to –340 °C·day were found at sites, where one plot was on the bare ground and the other, vegetated, such as C3, A1, A10, A12, and B11. The intra-plot distances in these sites were 14 and 16 m, except at C3 and B11 with those of 4 and 8 m, respectively.

For TDD, the intra-plot differences were slightly lower and for fewer sites (Fig. 11b). Sites D2 and A8, with both plots in swamp meadows, had an intra-plot difference in TDD of 320 and 180 °C·day, respectively. The distance between plots was only 2 m at site A8 and 8 m at site D2 (Table 1). Other significant differences of 350 and 270 °C·day were at sites A10 and C3, but with bare ground in one plot and swamp meadow in the other.

The fine-scale differences in the FDD and TDD are explained by the high heterogeneity of land surface covers and micro-topography rather than sensor errors. Higher FDD values are observed in the drier bare ground, while higher TDD values are observed in alpine swamp meadows with wet soils. Even in the same type of landcover, variations in microtopography and moisture affect FDD and TDD. In this area, Luo et al. (2020) reported an earlier start of ground freezing on the top of dry earth hummocks than in the adjacent moist depressions. Variations in soil moisture conditions explain the larger intra-plot difference in FDD at site A3 with both plots in alpine swamp meadow and on relatively flat terrain. In both 2019 and 2020, surface waters were present around plot A3A while 10 m further in plot A3B, surface waters were missing. However, these differences in moisture conditions did not affect the TDD, where differences were merely 52 °C·day.

On the Chalaping peat plateau, but over an area larger (3.5 km$^2$) than our local scale setting, FDD and TDD were calculated at 26 locations. FDD ranged between –931 and –1765 °C·day, and TDD, between 429 and 898 °C·day (Luo et al., 2020). At our local scale, we obtained a slightly larger variability of FDD and TDD with variations from –936 to –1952 °C.day and from 695 to 1144 °C.day (Fig. 11). Similar to their observations, the highest FDD were recorded in the plots in bare ground (exception in site A3), while the highest TDD, in vegetated plots. In addition, the FDD calculated at our sites are comparable with the ones derived from remote sensing thermal images for this area (Ran et al., 2021). The FDD from the vegetated sites in the HAYR are also comparable with the ones from the vegetated sites of discontinuous permafrost in Signy Island, Maritime Antarctica (Guglielmin et al., 2008). Nevertheless, the ones from the bare ground are lower than our bare ground sites. For the TDD, the values from HAYR are two to three times higher than those reported in Maritime Antarctica. Lower FDD and much lower TDD were reported from Livingston Island, Antarctica (Oliva et al., 2017; Hrbáček et al., 2020). Compared to Svalbard, the TDD from the HAYR are larger, while the FDD are lower (Christiansen et al., 2013).

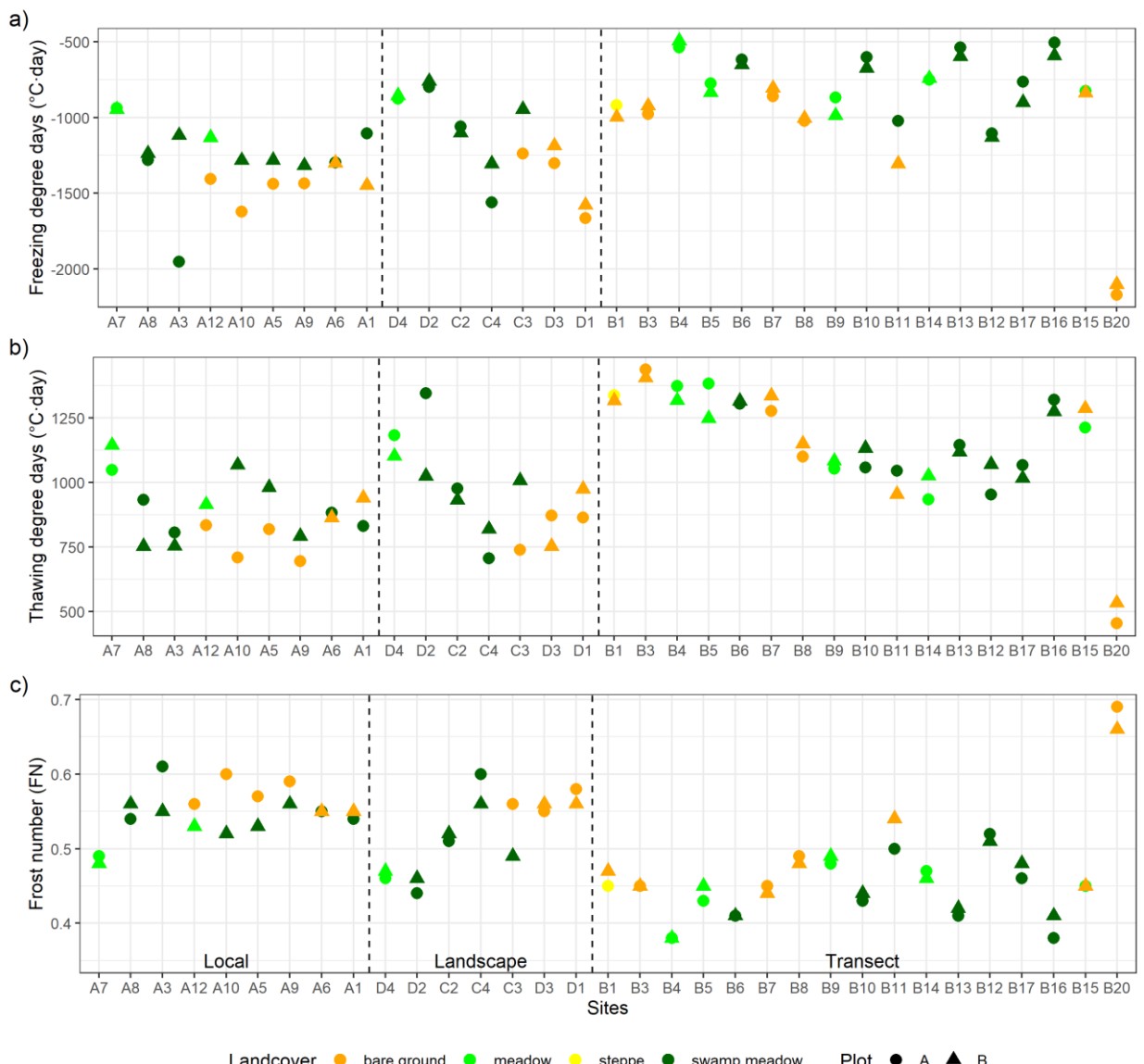

**Figure 11. Freezing degree days (a), thawing degree days (b), and the frost number (c) at the depth of the ground surface temperature, approximately 5 cm into the ground, for the period August 2019 to July 2020 in both plots (A and B) of each site in south-central HAYR. Notes: Local scale (2 km²), landscape scale (50 km²), and elevational transect (800 m difference).**

The values of FN were similar between the plots with differences below 0.05, with the exception at sites A10, A3, and C3 (Fig. 11c). The differences were up to 0.08 at site A10, where the bare ground was compared to swamp meadow at an intra-plot distance of 14 m. In terms of intra-site comparisons, FN ranged from 0.38 (B4 and B16) to 0.69 (B20), while for 88% of the sites, FN was between 0.4 and 0.6.

## 3.5 MAGST as a permafrost indicator

GST has been broadly used for understanding the thermal evolution of the SFG, active layer, and permafrost. Negative MAGST or low and stable GSTs in winter were interpreted as a confident indicator of the permafrost occurrence and distribution (Onaca et al., 2015; Vieira et al., 2017; Goncharova et al., 2019; Luo et al., 2019; Wani et al., 2020). On the QTP, a MAGST (including the maximum thermal offset of 0.79 °C) that is below or equal to 0 °C indicates the permafrost presence (Cao et al., 2019a). However, positive MAGSTs were reported in areas of permafrost boundary conditions from Scandinavia, the Alps, or Antarctica (Ishikawa, 2003; Ikeda, 2006; Etzelmüller et al., 2007; Colombo et al., 2020; Hrbáček et al., 2020). Therefore, due to the large spatiotemporal variability in GST, it was regarded a qualitative indicator to characterize the ground thermal state. For example, we can use MAGST values and their distribution to describe the ground thermal regime at a specific location and period, to identify the local cold and warm repartition, and to estimate the probability of permafrost occurrence (Bosson et al., 2015).

To test this supposition in the HAYR, we compared the MAGSTs from our sites with the ground temperature available from the boreholes along the elevational transect (Figs. 1 and 12). MAGST was compared with the multi-year (2010–2017) average temperatures at the depth of zero annual amplitude ($T_{ZAA}$) at 10 and 15 m depths (Luo et al., 2018b). The sites from the beginning of the elevational transect and up to 4400 m a. s. l. (B1 to B6) revealed a MAGST higher than 1 °C. They corresponded well with the $T_{ZAA}$ higher than 1 °C from borehole YNG-2 (4395 m a. s. l.), where SFG was present. The MAGST from the plots of the sites B7 and B8 varied from 0.1 to 1.1 °C and matched well with the $T_{ZAA}$ of –0.2 and –0.1 °C at the depths of 10 and 15 m from borehole YNG-1 at 4450 m a. s. l. At this site, permafrost was severely degraded and revealed a permafrost thickness of only 15 m. The next two sites, B9 and B10, showed a MAGST between 0.5 to 1.2 °C that relate with the warm permafrost from CLP-4 (4560 m a. s. l.) with a $T_{ZAA}$ of –0.6 °C (Fig. 12). These sites corresponded with the lower elevational limit of the discontinuous alpine permafrost from the northern flank of the Bayan Har Mountains identified at borehole YNG-1 at 4450 m a. s. l. (Luo et al., 2018b). The borehole CLP-3 with a $T_{ZAA}$ of –1.2 °C revealed stable permafrost and the lower limit of continuous elevational permafrost (4660 m a. s. l.). It agreed well with the MAGST of –0.42 °C from the vicinity of site B12. Sites at the landscape scale situated further away from the borehole, but at a similar elevational range (4600 to 4670 m a. s. l.), showed a MAGST between –2.2 and –1.2 °C (e.g., C3, D1, D3). Hence, these sites identified the presence of stable permafrost as well. However, site D4 at approximately the same elevational range (4600 m a. s. l.) revealed a MAGST of 0.7 °C which, as we previously saw, suggesting the presence of warm and degraded permafrost. This shows the high spatial variability of permafrost thaw over short distances that cannot be identified by the sparse borehole networks but can be reliably identified by monitoring the spatial variations in GST. In the absence of key boreholes, GST monitoring, combined with geophysical surveys, is more confident in detecting the presence or absence of permafrost (Hauck et al., 2004; Bosson et al., 2015; Onaca et al., 2015). It is a low-cost and non-invasive method that can cover even the most inaccessible and remote areas in the rough mountainous terrains. In terms of uncertainty, it is similar to geophysical methods, which could be complementary, but way more convenient in terms of logistics. On the other hand, borehole drilling and followed ground

temperature measurements are more precise but prohibitively expensive, inaccessible for rough terrains, consume large quantities of water, and are heavily invasive to the local ecosystems (Noetzli et al., 2021). Moreover, permafrost around the borehole is thawed during the drilling process and it requires several months to years to be able to record concluding ground temperatures (Kutasov and Eppelbaum, 2018). Furthermore, deep boreholes can increase the risk of gas escape to the surface with consequences to local populations, ecosystems, and the climate system (Gizatullin et al., 2023; Klotz et al., 2023).

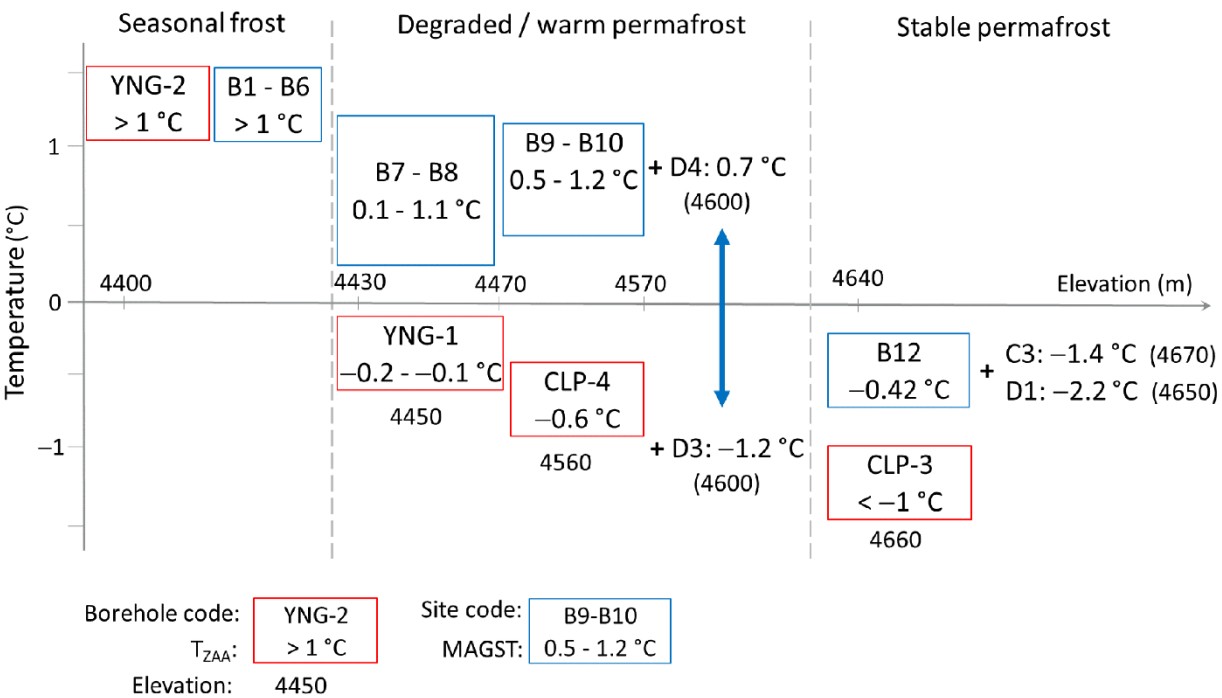

**Figure 12. Comparison of the mean annual ground surface temperature (MAGST) for the period August 2019 to July 2020 in south-central HAYR with the multi-year (2010-2017) average temperatures at the depth of zero annual amplitude (T$_{ZAA}$) at depths of 10 and 15 m. Notes: Boreholes data are from Luo et al. (2018b), T$_{ZZA}$ = Temperatures at the depth of zero annual amplitude (10 to 15 m).**

Considering that the criteria of FN > 0.5 and MAGST < 0 °C indicate the probable occurrence of permafrost (Heggem et al., 2006), at the local scale on the Chalaping plateau, only site A7 disregards these conditions (Figs. 9 and 11). This has been expected because this site is located between the two separate highway lines and has been disturbed by highway construction and operation (Șerban et al., 2023). On the landscape scale, these conditions are overpasses at sites D2 and D4 and by plot C3B, all located in the alpine meadow and swamp meadow. Most sites along the elevational transect suggest the permafrost absence or highly decayed along the highway. The exception is at site B20 from the top of the mountain and sites B11 and B12, which suggest permafrost conditions with an FN > 0.5 and MAGST < –1 °C. This agrees with previous studies that reported permafrost warming and deterioration along the engineering corridors on the QTP (Jin et al., 2000, 2006, 2008; Wu

et al., 2012; Luo et al., 2018c; Xing et al., 2023). At eight sites of monitoring permafrost along the Qinghai-Tibet Engineering Corridor, the annual mean soil temperature at 10 cm in depth ranged from –2.62 to –0.20 ºC (Zhao et al., 2021).

In zones of discontinuous and sporadic alpine permafrost, MAGST ranged from –3.7 to 2.4 °C in Southern Norway and from –2.7 to 3.1 °C in the Swiss Alps (Ikeda, 2006; Isaksen et al., 2011; Rödder and Kneisel, 2012). In the Swiss Alps, a MAGST of 1.5 and 2 °C was considered the boundary between the presence and absence of permafrost (Ikeda, 2006). At a site from the High Atlas Mountains (North Africa), a MAGST of 3.2 °C was interpreted as an indicator of the permafrost presence due to the low and stable winter temperature of –6.0 to –4.5 °C. The high MAGST was caused due to the high summer GSTs,

while the low and stable GSTs during winter were preserved under a thick and continuous snow cover at a site prone to snow accumulation (Vieira et al., 2017). Snow cover and its melting/ablation played key roles in the spatiotemporal variability of GST in many alpine areas (Apaloo et al., 2012; Rödder and Kneisel, 2012; Gisnås et al., 2014), but this might not be the case in the HAYR. Snow cover may be insignificant in influencing the GST in the HAYR due to its small thickness, discontinuity, and short-lived periods even though frequent snowfalls occur all year, it melts quickly under the strong solar radiation.

**4 Data availability**

The datasets are openly available from the National Tibetan Plateau/Third Pole Environment Data Center (https://dx.doi.org/10.11888/Cryos.tpdc.272945, Șerban and Jin, 2022).

**5 Conclusions**

A particular observational network for GST has been established in the HAYR covering various environmental conditions and

455 scales. GST was monitored from fine (distances up to 16 m) to local and landscape scales (2 and 50 km$^2$) and along an 800-m elevational transect. This analysis emphasized the large variability in GST over short distances (<16 m) in the HAYR over one hydrological year (2019–2020) under the influence of landcover conditions:

-The sites with similar landcover in both plots mainly presented an intra-plot difference of daily GST below 2 °C. The difference frequently exceeded 2 °C at sites where the bare ground was compared to vegetated plots. For some sites, from

460 autumn to spring, the difference increased to 4–5 °C for a duration of up to 15 days. This revealed a faster cooling of the bare ground under a strong coupling with air temperature due to the lack of a consistent layer of snow accumulation.

-The intra-plot variability in MAGST was < 0.5 °C (similar landcover in both plots), and increased to 1–2 °C when the bare ground was compared to vegetated plots. Higher variability was observed for the minimum annual GST, frequently exceeding 2–3 °C for sites with different landcover in the two plots.

-The annual FDD/TDD were quite similar between plots for most sites, while for some the differences were usually less than 350 °C·day. In the drier, bare ground, a higher FDD was observed, while in swamp meadows with a high moisture content, a higher TDD.

-The negative MAGST was associated with an FN > 0.5, indicating the probable presence of permafrost, and further validated by boreholes measurements. Sites with a MAGST < –1 °C corresponded well with the stable permafrost, while sites with a MAGST > 1 °C matched the areas of SFG. The sites with a MAGST that ranged between –1 and 1 °C agreed with the lower elevation limits of discontinuous permafrost on the northern flank of the Bayan Har Mountains.

This dataset at a high spatial resolution provides important data support for improving the upper boundary conditions in modelling approaches. It represents a useful input or validation dataset for permafrost and SFG models, as well as for reanalysis and remote sensing products. Particularly for the process-based numerical models, a dense network of GST can be used as input to better parameterize and calibrate the model. These will help to better represent the fluxes of energy exchange between the dynamic interaction of land and atmosphere due to the central position of GST in the Earth Critical Zone. Furthermore, a dense observational network helps to understand the effect of GST on the surface water cycles and vegetation development, soil organic carbon, and the rapid landscape changes in the HAYR. The observational network of GST on the HAYR is worth maintaining for long-term monitoring and intra-annual comparisons and could represent a model for other cold regions worldwide.

**Author contribution**

RȘ and HJ obtained funding and designed the study and the observational network. RȘ prepared the datasets and performed the analysis. MȘ plotted part of the figures and contributed to the field data acquisition. RȘ, MȘ, DL, QW, RH, XJ, and XL participated in the some of field work in establishing the study transects and boreholes in the HAYR and the ensued datasets for comparison analysis. RȘ wrote the manuscript and all authors revised the manuscript.

**Competing interests**

The authors declare that they have no conflict of interest.

**Acknowledgments**

This work was financially supported by the Autonomous Province of Bozen/Bolzano – Department for Innovation, Research and University in the frame of the Seal of Excellence Programme and the International Mobility for Researchers Programme (project TEMPLINK, Grant No. D55F20002520003 and project PERMAWAT, Grant No. 13585/2023); the sub-project of the Strategic Priority Research Program of Chinese Academy of Sciences (CAS) 'Impacts of changing permafrost hydrological processes in the Qilian Mountains on the water supplies and security' (Grant No. XDA20100103), and; the CAS Postdoctoral Program of President's International Fellowship Initiative-PIFI (2018PE0007). The authors are grateful for the support given by Mr. Fan Gao and Mr. Canjie Huang in the fieldwork.

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
