# Peer review of "An observational network of ground surface temperature under different landcover types on northeastern Qinghai-Tibet Plateau"

_Earth System Science Data, 2023_

## Referee Comment (RC3)

**General Comments**

This study conducted extensive ground surface temperature measurements in the Headwater Area of the Yellow River on the Qinghai-Tibet Plateau, providing abundant and valuable data for permafrost research in the QTP region. Based on the acquired data, the authors also conducted a detailed analysis and provided readers with insights into the possible applications of the current data in soil freeze/thaw research. The paper is well-organized, and the writing is clear and easily readable. However, there are still some issues that should be addressed before final publication.

**Specific comments:**

1.  In page 5, line 121, "local-scale sites are established in a flat peat plateau". In page 6, line 137, "some sites are covered by coarse gravel". Peat soils or gravel soils have distinct properties compared to fine mineral soils, and QTP is generally characterized by widespread gravel soil and generally low soil organic content (SOC). Therefore, more information should be included. For example, are there any measurements of topsoil organic content? Is the SOC in topsoil related to the intra-plot differences at sites all covered by same vegetation? Does the site covered with coarse gravel have any influence on the analysis results?

2.  The authors mentioned that this dataset can be useful as inputs or validations for permafrost and SFG models. Given the high spatial resolution of GST monitoring, providing information about soil texture or soil type at each site would be beneficial for model simulations and further analysis.

3.  The elevation cross-section is located on the northern side of the Bayan Har Mountains. Is there any information available regarding the slope and aspect of these locations? Does it have any impact on the results?

4.  One of the multiscale settings is the "fine scale," ranging from 2 to 16 meters. The authors stated that the fine-scale measurements were set for backup reasons and to identify the variations in GST. What were the criteria for setting two plots at each site? This scale is hardly matching the modeling or remote sensing applications. What are the potential applications of observations at the fine scale?

5.  The intra-plot differences at most sites are usually larger during the freeze-thaw transition period (Figs. 3-6), but at site B6 and B7, the same pattern is not observed and the differences are large throughout the entire year (Fig. 5). What are the possible reasons? Is there anything special about these two sites?

6.  Table 2. It's not surprising that R or R2 values are close to 1, but the RMSE or MAE provide more insightful information regarding GST variation at different scales. Additionally, investigating potential relationships between GST differences and environmental factors like elevation might be helpful. Including a figure to visualize these relationships could enhance the clarity of the analysis

**Technical corrections:**

Figure 1: add the lat/lons infromation, and adding a permafrost map as the background may be also helpful.

line 119-121: are these data from site CLP-1 or CLP-2?

Line 156: "Photographs were taken at each site and plot". I would suggest the authors add some photos to better present sites condition.

Line178: please briefly describe what AIC is.

Line 222: change "both" to "these two"

Figure 3-7: I would suggest the authors using same Y scale to better show the differences.

Figure 7d: the color difference between the two lines is minimal, making it difficult to distinguish the line representing "steppe."

Figure 8: I would suggest sorting the sites in transect by elevation to better present if there are elevation effects.

---

## Author Comment (AC1)

**Response to Referee 1**

The authors would like to thank the reviewer for the constructive feedbacks and kind advice, and the thorough assessment of the manuscript. Below, we are providing a point-to-point response to each comment: Reviewer comments are given in black, and our responses are given in blue. Additionally, we have included details of how we address these changes in the revised submission.

**General comments:**

This paper describes a ground surface temperature (GST) monitoring network established in a specific region of the Qinghai-Tibet Plateau. Temperature sensors were deployed across areas of varying surface characteristics to monitor changes in GST under different landcover conditions. The collected monitoring data is abundant and of reasonably high quality. The authors have conducted a thorough analysis of the acquired data, providing readers with a more in-depth understanding about the freeze-thaw state during that period. Overall, the English writing in this paper is clear and coherent, and the obtained data can serve as valuable input for modeling or validation of surface processes. However, there are still some issues that the authors should consider. I would be highly appreciated if the authors could address them.

Thank you for your kind summary! We have tried our best to address the raised issues as follows.

**Specific comments:**

1. What is the difference between the ground surface temperature (GST) mentioned in the paper and the land surface temperature (LST) commonly referred to in the remote sensing field, as well as soil temperature? Additionally, the description "topsoil temperature" in the data website provided by the authors raises questions about the physical meaning of the variables discussed in the paper. It is recommended that the authors either standardize their terminology or provide additional explanations within the text to ensure a clearer representation.

   The definition of the ground surface temperature (GST) is at lines 65-66: "GST is usually measured at approximately 5 cm into the ground but in literature, the GST depth was varying from 2 to 10 cm (Ferreira et al., 2017; Grünberg et al., 2020; Oliva et al., 2017; Onaca et al., 2015)".

   The land surface temperature (LST), measured either by remote sensing sensors or in-situ sensors, is the temperature at the surface of the landcover or on the top of the landcover. Thus, it is directly exposed to solar radiation and it is also known as the "skin temperature". We didn't add explanations of LST in the manuscript because it is not measured and does not represent the subject of this paper.

   We use the term "soil temperature" only when we refer to other studies with the temperature measured at different depths than the depth of GST at 5 cm. For example, at depths of 20 cm (line 306) or 10 cm (line 392).

In the data repository, we used the term "topsoil temperature" only in the title of the data more as a synonym of GST. In the summary of the dataset, we described that we refer to GST, which is measured at a depth of approximately 5 cm.

2. In the Introduction section, the authors mentioned that some scholars have already deployed GST monitoring networks in the northeastern part of the Qinghai-Tibet Plateau (e.g., Luo et al., 2020; Serban et al., 2023). What distinguishes the observational data in this study from those previous efforts? Perhaps the authors placed their monitoring network in mountainous regions? However, it seems that the data analysis by the authors did not include a specific analysis of mountainous characteristics. Despite some sections discussing elevation, the more unique features of mountainous regions such as three-dimensional structure and illumination conditions were not addressed.

Indeed, Luo et al. (2020) measured the GST but only at a few sites and on a small flat area (3.5 km$^2$) at Chalaping close to our sites from the local scale. This newly established monitoring network is covering a larger range in terms of elevational range and landcover types. However, not so much in terms of slope and aspect because the monitoring plots are located mostly in flat areas. The study area is on a high plateau with smoothed interfluves and peaks and the illumination conditions do not differ substantially.

Serban et al. (2023) analyzed the GST from this database but focused more on the intra-site comparison and detecting the environmental controls on GST variability. In that paper are included specific analysis and statistical tests regarding the environmental variables of topography (elevation and slope angle and aspect) and landcover types. The mountainous regions of the study area are described as well. We briefly referred to that in lines 83-84: "The variability of MAGST at other scales and their environmental controls have been assessed in detail by Serban et al. (2023)."

L306-308: "The intra-site MAGST variability has been mainly controlled by elevation and landcover types (Șerban et al., 2023), similar to observation from the Swiss Alps (Gubler et al., 2011)." has been replaced with "The intra-site MAGST variability has been mainly controlled by elevation and landcover types ( as is shown in Fig. 4 of Șerban et al., 2023), similar to observation from the Swiss Alps (Gubler et al., 2011). Slope and aspect do not play a relevant role because the monitoring plots are located mostly in flat areas (Șerban et al., 2023)."

L.97-98: "The study area is on a high plateau with smoothed interfluves and peaks and the illumination conditions do not differ substantially." has been added.

3. The title of the paper mentions a "multiscale observation network…" but typically, multiscale implies different sensor observation fields (e.g., ground stations, drones, satellites). However, in this study, all sensors used for observations are ground-based and have the same

observation field, with differences only in their placement. Additionally, it cannot be claimed that the sensors observed data at "local scale", "landscape scale", and "regional scale" because the instruments still provide sparse point observations and do not comprehensively cover an area. In summary, I am concerned about the validity and accuracy of the description "multiscale observation" in the paper.

If we refer to different sensor fields then is not a multiscale but we used this term more to have a way to spatially divide our study area. We also were inspired by similar repositories, such as:

https://data.tpdc.ac.cn/en/data/b6269aeb-8b44-4d03-b514-2c804c2cfc26/?q=soil%20temperature

We used the terms local and landscape scales, as well as the transect just to divide our study area based on its size, the differences in the environmental conditions, and the density of our sites. For example, the local scale represents an area of just 2 $km^2$ with homogeneous topographical conditions over a flat peat plateau with an elevational difference of only 18 m. In this area, GST is measured at 9 sites. Indeed, these measurements are point observations but because of their density and this relatively small area, we considered them representative of that area and the landcover type where they are placed.

To avoid confusion, we removed the term "multiscale". The following changes have been made:

L.1-2: "Multiscale observation network of ground surface temperature under different landcover types on NE Qinghai-Tibet Plateau" has been replaced with "An observational network of ground surface temperature under different landcover types on northeastern Qinghai-Tibet Plateau"

L.459: "The multiscale observational network" has been replaced with "The observational network"

4. The authors mentioned that some sensors were malfunctioning. What is the current status of these sensors? Are they now operational, or are they still not functioning correctly? Is there a possibility of acquiring more comprehensive observational data in the future?

The sensors that were malfunctioning were replaced with new ones. It is planned to visit again the sites this October to check their status and download the data for the second time after a long-term measurement due to the COVID-19 epidemic.

5. Page 6, line 154. The authors mentioned a data collection interval of 3 hours for ground observations. Does this mean that data is recorded once every 3 hours, or is it recorded

multiple times and then averaged using a specific algorithm? I suggest providing a brief explanation in the paper for clarity.

The data is recorded once every 3 hours, not multiple measurements and averaged. The following changes have been made:

L.162-163: "…at a 3 h interval" has been replaced with "and the temperature was recorded once every three hours."

6. Page 9, line 206. How were the 165 "biased" data points mentioned in the paper determined? Were they identified through manual inspection or using a specific criterion (e.g., three times the standard deviation screening)?

The biased values were determined through manual inspection, plotting the timeseries, and checking the minima and maxima. These values were easily detected because represent extreme values, such as –41 ºC or 87 ºC. The 165 biased values are described in the following paragraph:

L.219-223: "From these, the most severe one was found in plot A3A, with a period of 10 days from 1 to 19 September 2019, with 151 measurements blocked at –41 ºC. In addition, there were another 13 erroneous measurements with temperatures of –41, –39.5, and 87 ºC on 23 and 26 February 2023. The sensor from plot B16B had only one wrong measurement of –7.7 ºC on 17 October 2019, while the other temperature readings during that period ranged from 0.1 to 2.6 ºC."

7. Page 14, line 266. Why is it that a 14-meter distance can observe larger GST differences for the same type of landcover type?

When we said larger GST differences at intra-plot distances of 14 m, we were referring to all sites and these differences mainly occurred when vegetated plots were compared to the bare ground.

There were only two sites with both plots in alpine swamp meadow with several days of larger intra-plot GST differences. From them, only site C4 had an intra-plot distance of 14 m, while site D2 had an intra-plot distance of 8 m. Timeseries of these sites are represented in Figs. 7 e and 7f. This was also observed for the mean annual ground surface temperature (MAGST) where the intra-plot differences were below 0.5 °C, especially for sites with the same landcover in both plots.

Even though we observed these differences at 14-m distances, especially for comparing bare ground to vegetated sites, the statistical tests did not show a significant influence of the intra-plot distance (Please see Serban et al., 2023).

These observations were summarized in lines 291-294: "The intra-plot variability of MAGST was mainly below 0.5 °C, especially for sites with the same landcover in both plots (Fig. 8a). Variations between 1 and 2 °C were observed at the sites where the bare ground was compared to vegetated plots (A10, C3, and B11). This variability was observed mainly at all analyzed distances between plots and according to the analysis of variance and linear regression, it was insignificantly influenced by distance (Șerban et al., 2023)."

For the sites with swamp meadow in both plots and with several days/periods of higher differences in GST (e.g., C4) we assumed the cause is the variability in moisture content. Some plots had a higher moisture content and were oversaturated, even with the presence of surface water around them. While in the nearby plots, soil moisture content was lower without the presence of surface water. Like comparing to the drier vegetation plots from the alpine meadow (L.310-311) the evaporative cooling and the variability of the moisture content may cause a higher thermal offset.

More detailed explanations we provided in Șerban et al. (2023), such as:

"The high soil water content from oversaturated swamp meadows assures a high heat capacity and thermal conductivity of the soils than those in the drier meadows…."

"Rich soil moisture contents or presence of surface water body will retard the ground freezing or thawing due to the huge fusion heat of phase change either ice/water (melting), water/vapor (evaporation), or ice/vapor (sublimation). Freeze-up of icy soils in the active layer or in lakes/wetlands will release heat more efficiently in winter. In the meantime, lower thermal conductivity of dry, thawed/unfrozen organic soils and higher thermal conductivity of ice-rich frozen soils result in higher thermal offsets. At the same time, intense evapotranspiration will cool the ground more effectively in summer along water surfaces that may also absorb more heat, but they are not in the same order of magnitude."

"A reduction in soil temperature variations was also observed in the Arctic caused by the higher thermal conductivity of wet soils and the high heat capacity of water (Aalto et al., 2013)."

"…higher moisture boosted evapotranspiration, which in turn lowered GST (Aalto et al., 2013)."

"Detailed *in-situ* observations on snow cover and soil conditions (texture, moisture, and organic content) are needed to better understand the controls of GST in the HAYR. These soil properties are strongly influencing the soil thermal conductivity, heat capacity, and hydraulic conductivity that affects the soil freeze/thaw processes (Jiang et al., 2020) and subsequently the GST variability."

8. Page 17, line 290. Although the authors have provided some explanations regarding the relationship between MAGST and elevation, it might be more intuitive to include a graphical representation of the MAGST and elevation relationship.

Indeed, the influence of elevation on GST spatial variability has been detailed assessed, including a graphical representation of the decrease of GST with elevation (please see Figure 4 from Șerban et al., 2023). In this data paper, we avoided repeating the same analysis and we focused more on the intra-plot variability of GST, and added only a reference:

L306-307: "The intra-site MAGST variability has been mainly controlled by elevation and landcover types ( as is shown in Fig. 4 of Șerban et al., 2023), similar to observation from the Swiss Alps (Gubler et al., 2011)."

Please also see the reply to the comment number two.

9. Page 19, lines 341-348. While it is understandable that the authors compare the results of FDD calculations with previous satellite-based calculations, is it meaningful to compare the results with very distant regions like Antarctica or other islands (especially when the timeframes are not consistent)?

We considered that besides comparing the FDD and TDD to other works on the QTP, it is worth to also compare it to other permafrost environments for a global overview of these indices. QTP is part of the "Third Pole" region, thus a comparison to the Arctic and Antarctica areas is deemed necessary to better fit with the special issue "*Extreme environment datasets for the three poles*" to each the manuscript is submitted.

10. Page 20, lines 374-376. While the authors mention that GST monitoring can provide a better assessment of the presence or absence of permafrost, they also note the high spatial variability of permafrost thaw. In my view, for an accurate determination of permafrost status, even when using GST as an indicator, a highly dense sensor network would be necessary, which does not seem to be currently feasible. Therefore, the authors need to further explain why they chose GST monitoring for assessing permafrost status over other methods such as borehole measurements (considering factors like cost, convenience, data uncertainty, etc.).

Boreholes are more precise but expensive and invasive. The heavy machine destroys the grasslands and ecosystems, and the drilling requires a lot of water. Drilling also affects the ground at deeper depths, it thaws the permafrost and it requires several years to become stable again and to record concluding measurements. GST monitoring is a non-invasive method, low cost, and with faster results/measurements. The following changes have been made:

L.396-403: "It is a low-cost and non-invasive method that can cover even the most inaccessible and remote areas in the rough mountainous terrains. In terms of uncertainty, it is

similar to geophysical methods, which could be complementary, but way more convenient in terms of logistics. On the other hand, borehole drilling and followed ground temperature measurements are more precise but prohibitively expensive, inaccessible for rough terrains, consume large quantities of water, and are heavily invasive to the local ecosystems (Noetzli et al., 2021). Moreover, permafrost around the borehole is thawed during the drilling process and it requires several months to years to be able to record concluding ground temperatures (Kutasov and Eppelbaum, 2018). Furthermore, deep boreholes can increase the risk of gas escape to the surface with consequences to local populations, ecosystems, and the climate system (Gizatullin et al., 2023; Klotz et al., 2023)." has been added.

11. Page 22, line 425. While the authors mention the potential significance of this dataset for improving modeling methods, the entire paper analyzes the relationship between GST and freeze-thaw without specifying the advantages of higher spatial resolution GST monitoring data for model improvement (compared to using satellite data). Considering that large-scale snow and ice state analysis typically relies on satellite observations, is there a genuine necessity for such dense sensor deployment?

Yes, a dense sensor network will help to validate at a higher spatial resolution the satellite data products (e.g., land surface temperatures, snow distribution, reanalysis climatic grid datasets) and the models of permafrost spatial distribution.

These products are still too coarse to reproduce this high spatial variability of the ground temperature as was observed in the monitoring of GST. Moreover, the actual permafrost models rely on these coarse data as inputs and are not able to detect the fine scale patterns of permafrost thawing and thermal status. The increasing availability of high resolution remote-sensing derived products need an increasing variability of accurate datasets for their validation. It is important to highlight that every remote-sensing derived product is the result of a complex modelling chain of the signal, which often requires strong assumptions on the physics of the soil surface that require validation data.

Moreover, it is important to underline that remote sensing approaches measure LST and not GST. Deriving GST from LST requires a physical or statistical modelling approach (Endrizzi et al., 2014). Particularly for the process-based numerical models, a dense network of GST can be used as input to better parameterize and calibrate the model. These will improve the upper boundary conditions of the ground profile and will help to better represent the fluxes of energy exchange between the dynamic interaction of land and atmosphere. The GST is the key parameter controlling all the bio-physical processes at the land-atmosphere boundary due to its central position in the Earth Critical Zone.

To better elaborate the necessity of a dense observational network of GST with emphasis on its usefulness for modelling approaches, the following changes have been made:

L.455-457: "Particularly for the process-based numerical models, a dense network of GST can be used as input to better parameterize and calibrate the model. These will help to better

represent the fluxes of energy exchange between the dynamic interaction of land and atmosphere due to the central position of GST in the Earth Critical Zone." has been added.

L.457-458: "Furthermore, is helpful for understanding the effect…" has been replaced with "Furthermore, a dense observational network helps to understand the effect…"

**Technical corrections:**

1. Page 2, line 46. The term "permafrost areal extents" is also a component of "model accuracies", so there is no need to repeat it.

   L.47-48: "…but with significant differences in permafrost areal extents and in model accuracies…" has been replaced with "…but with significant across-model differences in model accuracies …"

2. In Figure 1, there is an issue with the legend labels. "locale" should be "local". Additionally, please confirm whether "Qingshui'he" should be "Qingshuihe".

   Thank you for notifying that. "locale" has been replaced with "local".

   "Qingshui'he" is the correct term and has also been used more often in previous publications.

3. Page 9, line 194. Are the mentioned four failed sensors included among the previously mentioned 11 sensors, or are they an additional set of four sensors?

   They are included. Thank you for pointing out this unclarity. The following changes have been made:

   L.205-207: "Four sensors had become malfunctioned and have not recorded any measurements, while three sensors stopped recording the measurements after seven months." has been replaced with "Among the 11 malfunctioned sensors, four sensors had become malfunctioned without any recorded measurements, while three sensors stopped recording the measurements seven months after installations. ."

4. Page 10. In the title of Figure 3, there is no need to repeatedly provide the full term of "GST".

   The full term of "GST" has been removed from the title of Figure 3, as well as from Figures 4, 5, 6, and 7.

5. Page 12. I suggest adding a legend to Figure 5.

   The legend has been added to Figure 5. In the legend in Figures 3, 4, and 6, the "swamp meadow / bare ground" has been replaced with "vegetation / bare ground".

6. Page 17, line 318. "… TDD of 320 m and 180 ℃ day", remove "m".

   L.336: "… TDD of 320 m and 180 °C·day, respectively …" has been replaced with "…TDD of 320 and 180 °C·day, respectively …"

7. Page 18, line 328. When the authors mention "… most of the sites", I suggest giving the exact percent.

   L.345-346: "… while for most of the sites, …" has been replaced with "…while for 88% of the sites,  …"

References:

Aalto, J., Le Roux, P. C., and Luoto, M.: Vegetation mediates soil temperature and moisture in arctic-alpine environments, Arctic, Antarct. Alp. Res., 45, 429–439, https://doi.org/10.1657/1938-4246-45.4.429, 2013.

Endrizzi, S., Gruber, S., Dall'Amico, M., Rigon, R.: GEOtop 2.0: Simulating the combined energy and water balance at and below the land surface accounting for soil freezing, snow cover and terrain effects. Geosci. Model Dev. 7, 2831–2857. https://doi.org/10.5194/gmd-7-2831-2014, 2014.

Jiang, H., Zheng, G., Yi, Y., Chen, D., Zhang, W., Yang, K., Miller, C.E.: Progress and challenges in studying regional permafrost in the Tibetan Plateau using satellite remote sensing and models. Front. Earth Sci. 8, 1–17. https://doi.org/10.3389/ feart.2020.560403, 2020.

Șerban, R. D., Bertoldi, G., Jin, H., Șerban, M., Luo, D., and Li, X.: Spatial variations in ground surface temperature at various scales on the northeastern Qinghai-Tibet Plateau, China, Catena, 222, 106811, https://doi.org/10.1016/j.catena.2022.106811, 2023.

---

## Author Comment (AC2)

**Response to Referee 2**

The authors would like to thank the reviewer for the constructive feedback and kind advice, and the thorough assessment of the manuscript. Below, we are providing a point-to-point response to each comment: Reviewer comments are given in black, and our responses are given in blue. Additionally, we have included details of how we address these changes in the revised submission.

The authors provided a valuable dataset of GST observations at various spatial scales in the Headwater Area of the Yellow River (HAYR). GST datasets were collected at 39 sites between 2019 and 2020. The authors showed how the measurements could be used for permafrost research.

Thank you for your kind summary! We have tried our best to address the raised issues as follows.

**General Comments**

(1) Overall picture

While the authors provide a very detailed comparison of GST at different scales, this study generally lacks an overall picture. An easy way to do this would be to examine the lapse rate of MAGST. There should be a new figure with the x-axis representing elevation and the y-axis representing MAGST. You could even use different colors to represent vegetation cover.

Indeed, as suggested by Referee 1, the influence of elevation on GST spatial variability has been detailed and assessed, including a graphical representation of the decrease of GST with elevation (please see Figure 4 from Şerban et al., 2023). In that figure, the elevation is exactly represented on the x-axis, MAGST on the y-axis, and landcover with different colors. Additionally, the regression lines are added to indicate that the more significant decrease of MAGST is visible in bare ground than in vegetated sites. The lapse rate has been calculated as well:

"The MAGST in the study area declines with a vertical lapse rate of –3.9 °C/km for alpine meadows, –7.3 °C/km for bare grounds, and –9.4 °C/km for alpine swamp meadows. Considering all landcover types together, MAGST lowers at a vertical lapse rate of –6.8 °C/km." (Şerban et al., 2023).

In this data paper, we avoided repeating the same analysis and we focused more on the intra-plot variability of GST. We only briefly mentioned:

L83-84: "The variability of MAGST at other scales and their environmental controls have been assessed in detail by Serban et al. (2023)."

L306-308: "The intra-site MAGST variability has been mainly controlled by elevation and landcover types (Șerban et al., 2023), similar to observation from the Swiss Alps (Gubler et al., 2011)." has been replaced with "The intra-site MAGST variability has been mainly controlled by elevation and landcover types ( as is shown in Fig. 4 of Șerban et al., 2023), similar to observation from the Swiss Alps (Gubler et al., 2011). Slope and aspect do not play a relevant role because the monitoring plots are located mostly in flat areas (Șerban et al., 2023)."

(2) Permafrost borehole temperature datasets

A borehole temperature measurement from Luo et al., 2018 was used to determine whether permafrost was present. As an additional dataset, I suggest authors make the borehole temperature measurements public open.

That dataset of borehole temperature is not openly available at this moment but is planned to be published in the near future. However, in our comparisons, we only used the average values from Table 1 from Luo et al., 2018 and not the full dataset. Therefore, we only cite the respective paper.

(3) Review of GST measurements

In the CMA monitoring network, GST has been measured since the 1950s on the QTP and even the entire country. In spite of this, the measurement algorithm is inconsistent, making direct use of the dataset problematic (see Cui et al., 2020, Cao et al., 2023). Therefore, the datasets here are valuable. It would be helpful if you reviewed the measurement algorithms and clarified your significance.

Although this work is not focused on reviewing older measurements of GST from other networks, we further emphasized the importance of this dataset recorded through automatic measurements as suggested. The following changes have been made:

L62-65: "Although the GST started to be manually measured since the 1950s through the network of the China Meteorological Administration, these earlier measurements were inconsistent with the recent automatic measurements. Furthermore, the manual protocol of historical measurements was highly biased by the presence of snow cover (Cao et al., 2023; Cui et al., 2020)." has been added.

**Specific Comments**

L37: ...approximate or about 55%.

L39: "because 55%" has been replaced with "because about 55%"

L39: Cao et al., 2019 PPP reported the permafrost zonation index map based on a statistical model and various measurements. Please consider citing here.

L40: "and 41% by permafrost (Zou et al., 2017; Cao et al., 2022)" has been replaced with "and 40 to 46% by permafrost (Zou et al., 2017; Cao et al., 2019b, 2022)."

L44: Cao et al., 2018, JGR-Atmospheres reported the permafrost changes over the Northeastern QTP.

In lines 44 and 45, we speak about the models that predict the spatial distribution and the future evolution of the permafrost at the regional level of the entire QTP. The suggested paper analyzed in-situ observations from boreholes in a small area from northeastern QTP and it does not match with the context of this paragraph.

L45-46: "Earth system models predicted that permafrost thicker than 10 m covers 36% of the QTP and permafrost thickness will continue to decrease at rates of up to 21 cm per year under various climate change scenarios (Zhao et al., 2022)".

However, the suggested paper also reports values of MAGST observations from another mountain range from the northeastern QTP and is more suitable for comparison in the results and discussions section. The following changes have been made:

L322-323: "MAGST variations of up to 3 °C were also reported from the Qilian Mountains on the northeastern QTP (Cao et al., 2018)." has been added.

L56: Cao et al., 2020 TC (Table 1) reported how the MAGST combined with thermal offset can be used as an indicator for permafrost presence/absence.

Thank you very much for this suggestion. Indeed, very useful thresholding was determined for the QTP. The following changes have been made:

L57-58: "… delineate the distribution of SFG and permafrost (Rödder and Kneisel, 2012; Vieira et al., 2017; Luo et al., 2019; Wani et al., 2020; Serban et al., 2021; Jiao et al., 2023)(Cao et al., 2019; Jiao et al., 2023; Luo et al., 2019; Rödder and Kneisel, 2012; Serban et al., 2021; Vieira et

al., 2017; Wani et al., 2020).” has been replaced with “… delineate the distribution of SFG and permafrost (Rödder and Kneisel, 2012; Vieira et al., 2017; Luo et al., 2019; Cao et al., 2019a; Wani et al., 2020; Serban et al., 2021; Jiao et al., 2023).”

L369-370: “On the QTP, a MAGST (including the maximum thermal offset of 0.79 °C) that is below or equal to 0 °C indicates the permafrost presence (Cao et al., 2019a).” has been added.

L104: Please clrify the landcover and microtopography information here.

L111-115: “Therefore, GST has been monitored in different landcover types, such as the alpine steppe, meadow, swamp meadows, and bare grounds. In terms of microtopography, GST is monitored mostly on flat terrains but also in disturbed grounds by highway construction, thermokarst depressions, between thermokarst ponds, earth hummocks, and near gullies..” has been added.

L121-122: “Sites are placed in the proximity of both sides of the highway in different landcover types, such as the alpine steppe, meadow, swamp meadow, and bare ground.” has been replaced with “Sites are placed in the proximity of both sides of the highway in different landcover types.”

L132-133: “Differentiation is caused by micro-topography and landcover variety because sites are placed in alpine meadows, swamp meadows, bare grounds, disturbed grounds by highway construction, thermokarst depressions, between thermokarst ponds, and near gullies. The linear distance between sites is ranging from 70 to 465 m, with an average of 275 m.” has been replaced with “Differentiation is caused by micro-topography and landcover variety, while the linear distance between sites ranges from 70 to 465 m, with an average of 275 m.”

L135: *“…for some sites…”*, please give the number of sites which have similar landcover.

L142-143: “For some sites, …” has been replaced with “For 26 sites, ….”

L170: change *larger* to greater

L178: “An FN larger than 0.5” has been replaced with “An FN greater than 0.5”

L173: The principle behind SO and TO is the effects of vegetation cover, and soil properties (soil organic content, soil moisture). Please clarify here.

L182-185: “The surface offset is driven by snow cover and solar radiation and controlled by topography and vegetation. Thermal offset is mainly controlled by heat transfer and influenced by

different soil thermal conductivities in the frozen and thawed states determined by soil properties, such as soil texture and soil moisture, and organic contents (Smith and Riseborough, 2002; Wani et al., 2020)." has been added.

L193: change "*delete*" to remove

L204: "to detect measurement errors" has been replaced with "to detect and remove measurement errors"

L231: "*Differences larger than 2.5 °C were observed mainly at the sites at elevations above 4600 m a. s. l., regardless of the landcover types in the plots.*" why?

These larger differences at higher elevations could be related to the temperature inversion observed on the elevational transect (Șerban et al., 2023). While these temperature inversions showed seasonality, being more visible in spring and especially in winter, they could also indicate a diurnal variability as these high daily differences appear predominantly in autumn, winter, and spring. The following changes have been made:

L248-252: "The larger intra-plot difference in daily GST at higher elevations may be related to the temperature inversion observed on the elevational transect. These temperature inversions showed seasonality, being more visible in spring and especially in winter (Șerban et al., 2023). However, they could also indicate a diurnal variability caused by the strong radiation cooling under dry conditions and the local air circulation. The spatial differences in the reduction of plant species and root biomass could also increase the GST due to the decrease of evapotranspiration (Du et al., 2007)." has been added.

Fig.1: Please add the specific distance for each scale in the legend.

 The specific distance for each scale has been added in the legend.

**References**

Cao, B., Zhang, T., Peng, X., Mu, C., Wang, Q., Zheng, L., Wang, K., & Zhong, X. (2018). Thermal Characteristics and Recent Changes of Permafrost in the Upper Reaches of the Heihe River Basin, Western China. *Journal of Geophysical Research: Atmospheres*, *123*(15), 7935–7949. https://doi.org/10.1029/2018JD028442

Cao, B., Zhang, T., Wu, Q., Sheng, Y., Zhao, L., & Zou, D. (2019). Permafrost zonation index map and statistics over the Qinghai-Tibet Plateau based on field evidence. Permafrost and Periglacial Processes, 30(3), 178–194. https://doi.org/10.1002/ppp.2006

Cao, B., Zhang, T., Wu, Q., Sheng, Y., Zhao, L., & Zou, D. (2019). Brief communication: Evaluation and inter-comparisons of Qinghai–Tibet Plateau permafrost maps based on a new inventory of field evidence. *The Cryosphere*, *13*(2), 511–519. https://doi.org/10.5194/tc-13-511-2019

Cao, B., Wang, S., Hao, J., Sun, W., & Zhang, K. (2023). Inconsistency and correction of manually observed ground surface temperatures over snow-covered regions. *Agricultural and Forest Meteorology*, *338*(November 2022), 109518. https://doi.org/10.1016/j.agrformet.2023.109518

Cui, Y., Xu, W., Zhou, Z., Zhao, C., Ding, Y., Ao, X., & Zhou, X. (2020). Bias Analysis and Correction of Ground Surface Temperature Observations across China. Journal of Meteorological Research, 34(6), 1324–1334. https://doi.org/10.1007/s13351-020-0031-9

Luo, D., Jin, H., Jin, X., He, R., Li, X., Muskett, R. R., Marchenko, S. S., & Romanovsky, V. E. (2018). Elevation-dependent thermal regime and dynamics of frozen ground in the Bayan Har Mountains, northeastern Qinghai-Tibet Plateau, southwest China. Permafrost and Periglacial Processes, 29(4), 257–270. https://doi.org/10.1002/ppp.1988

Du, M., Kawashima, S., Yonemura, S., Yamada, T., Zhang, X., Liu, J., Li, Y., Gu, S., and Tang, Y.: Temperature distribution in the high mountain regions on the Tibetan Plateau - Measurement and simulation, in: MODSIM07 - Land, Water and Environmental Management: Integrated Systems for Sustainability, Proceedings, 2146–2152, 2007.

Luo, D., Jin, H., Jin, X., He, R., Li, X., Muskett, R. R., Marchenko, S. S., and Romanovsky, V. E.: Elevation-dependent thermal regime and dynamics of frozen ground in the Bayan Har Mountains, northeastern Qinghai-Tibet Plateau, southwest China, Permafr. Periglac. Process., 29, 257–270, https://doi.org/10.1002/ppp.1988, 2018b.

Șerban, R. D., Bertoldi, G., Jin, H., Șerban, M., Luo, D., and Li, X.: Spatial variations in ground surface temperature at various scales on the northeastern Qinghai-Tibet Plateau, China, Catena, 222, 106811, https://doi.org/10.1016/j.catena.2022.106811, 2023.

Smith, M. W. and Riseborough, D. W.: Climate and the limits of permafrost: a zonal analysis, Permafr. Periglac. Process., 13, 1–15, https://doi.org/10.1002/ppp.410, 2002.

Wani, J. M., Thayyen, R. J., Gruber, S., Ojha, C. S. P., and Stumm, D.: Single-year thermal regime and inferred permafrost occurrence in the upper Ganglass catchment of the cold-arid Himalaya, Ladakh, India, Sci. Total Environ., 703, 134631, https://doi.org/10.1016/j.scitotenv.2019.134631, 2020.

---

## Author Comment (AC3)

**Response to Referee 3**

The authors would like to thank the reviewer for the constructive feedback, and the thorough assessment of the manuscript. Below, we are providing a point-to-point response to each comment: Reviewer comments are given in black, and our responses are given in blue. Additionally, we have included details of how we address these changes in the revised submission.

**General Comments**

This study conducted extensive ground surface temperature measurements in the Headwater Area of the Yellow River on the Qinghai-Tibet Plateau, providing abundant and valuable data for permafrost research in the QTP region. Based on the acquired data, the authors also conducted a detailed analysis and provided readers with insights into the possible applications of the current data in soil freeze/thaw research. The paper is well-organized, and the writing is clear and easily readable. However, there are still some issues that should be addressed before final publication.

Thank you for your kind summary and appreciation!

**Specific comments:**

1. In page 5, line 121, "local-scale sites are established in a flat peat plateau". In page 6, line 137, "some sites are covered by coarse gravel". Peat soils or gravel soils have distinct properties compared to fine mineral soils, and QTP is generally characterized by widespread gravel soil and generally low soil organic content (SOC). Therefore, more information should be included. For example, are there any measurements of topsoil organic content? Is

the SOC in topsoil related to the intra-plot differences at sites all covered

by same vegetation? Does the site covered with coarse gravel have any

influence on the analysis results?

We do not have measurements of topsoil organic content but we have grain-size analysis and measurements on the water content for 11 sites situated in alpine meadow and bare ground. Unfortunately, we do not have soil samples from both plots of a site and only from one plot per site. Thus, we cannot consider the soil texture in the fine-scale analysis of the intra-plot differences regarding the ground surface temperature (GST). However, this suggestion is an interesting point that we will consider in the next field campaign, and will collect additional soil samples from key sites and plots to perform more analysis on soil characteristics.

Regarding the last question, we performed in the past grain-size analysis on the available soil samples and added more information in the manuscript. The following changes have been made:

L168-169: "From 11 plots, soil samples were collected (Table 1) for grain size and water content analysis. Samples were weighed before and after being dried at 105 °C for 16 hours to determine the water content. The coarse texture represented by gravel (> 2 mm) was quantified by sieving, while the fine texture (< 2 mm) representing sand, silt, and clay was measured with a Malvern Mastersizer-2000 laser diffraction particle size analyzer." has been added.

L304-311: "To better understand the variability of MAGST under the influence of landcover, even under the same landcover type, the MAGST was compared with the soil texture and soil water content (Fig. 10, Table 3). All the samples from the local and landscape scales collected from bare ground and with a fine texture of above 75% (Fig. 10a) revealed a low MAGST ranging between –2.2 and –1.2 °C (Fig. 10b). The samples from the lower part of the elevational transect (up to 4432 m) revealed a MAGST between 1.3 and 1.7 °C, regardless of landcover type (meadow or bare ground) or texture. Three of them had a fine texture between 70 and 98%, except plot B3A with the lowest fine texture of 59%. Only plot B8A from bare ground with a fine texture of 73% had the lowest MAGST of 0.2 °C among the elevational sites sampled for soil texture. However, the elevation of plot B8A is still relatively low, only 4473 m a. s. l." has been added.

L314-317: "Figure 10. Textural classification of the soil samples from selected plots (a) and their corresponding mean annual ground surface temperature (MAGST) for the period August 2019 to July 2020 in south-central HAYR (b). Notes: Coarse soil texture is represented by gravel while fine soil texture by sand, silt, and clay, with the threshold between them set at 2 mm in diameter." has been added.

L320-321: "Table 3. Grain size distribution and water content of the soil samples from selected plots." has been added.

2. The authors mentioned that this dataset can be useful as inputs or

validations for permafrost and SFG models. Given the high spatial

resolution of GST monitoring, providing information about soil texture or

soil type at each site would be beneficial for model simulations and further

analysis.

Indeed, the soil type and texture are very useful for a better parameterization of physical models. Unfortunately, only from 11 plots were carried out analysis on soil texture as we detailed in the response to the previous comment. Results of this grain-size analysis were added in Table 3 and Figure 10. Because the soil analysis does not cover all the sites we emphasized only the utility of the GST dataset for validating other models or remote sensing products. Furthermore,  the GST dataset can be also used to improve the upper boundary conditions in simulations of soil temperature or permafrost distribution by using physical models.

3. The elevation cross-section is located on the northern side of the Bayan Har

Mountains. Is there any information available regarding the slope and

aspect of these locations? Does it have any impact on the results?

The information regarding the slope and aspect of the monitoring sites are detailed in Table 1 from Șerban et al., 2023. In the respective work, statistical tests (Pearson correlation, linear models, and analysis of variance – ANOVA) were performed to identify the environmental controls on GST variability in this area. The results showed that slope and aspect did not have any statistically significant influence on the GST variability and only the landcover and elevation. In fact, in this region the topography is relatively smooth. In this data paper, we avoided repeating the same information and analysis and we focused more on the intra-plot variability of GST and added only a reference.

The following changes have been made:

L323-325: "The intra-site MAGST variability has been mainly controlled by elevation and landcover types (Șerban et al., 2023), similar to observation from the Swiss Alps (Gubler et al., 2011)." has been replaced with "The intra-site MAGST variability has been mainly controlled by elevation and landcover types (as is shown in Figs. 4 and 8 of Șerban et al., 2023), similar to observation from the Swiss Alps (Gubler et al., 2011). Slope angles and aspects do not play a relevant role because the monitoring plots are located mostly in flat areas (Șerban et al., 2023)."

4. One of the multiscale settings is the "fine scale," ranging from 2 to 16

meters. The authors stated that the fine-scale measurements were set for

backup reasons and to identify the variations in GST. What were the criteria

for setting two plots at each site? This scale is hardly matching the modeling

or remote sensing applications. What are the potential applications of

observations at the fine scale?

Indeed, in line 140 we said "This was done due to backup reasons and to identify the variations in GST at a fine scale." The main reason was to identify the variability of GST across various short distances ranging from 2 to 16 m under similar topographical conditions and differences only in terms of the landcover type. For several sites, even the landcover type was similar and only the intra-plot distance was different. Complementary, to the fine scale comparisons of the GST evolution in this data paper, in Șerban et al., 2023 were emphasized the intra-plot differences in MAGST according to the intra-plot distance. The differences in MAGST were higher when the intra-plot distance was above 8 m. However, there was no statistical significance probably due to the low number of samples for the statistical test. However, the intra-plot difference in MAGST was more clear when the plots from bare ground were compared to vegetated ones.

The second reason for two plots in each site was for backup reasons because as can be seen in Table 1 that in several plots the sensors failed to acquire a complete timeseries of data. Therefore, the data available in the other plot was used for the intra-sites comparison, and therefore in all sites, there was at least one plot available with a complete timeseries of GST. An exception was only in site B18 where the data is not complete in both plots. Details on the missing data and incomplete timeseries are in subchapter 3.1 Data quality check (L.203-224).

The scale does not match the spatial resolution of the most common free remote sensing products from satellite images but matches the special resolution of the thermal images from unmanned aerial vehicles (UAVs) and airplane images. This scale of 2 to 16 m is relevant for high-resolution modeling and remote sensing products. For example, the spatial resolution of both optical and thermal bands is between 0.3 and 4 m for many commercial high-resolution sensors, such as WordView, GeoEye, KOMPSAT, Quickbird, Pléiades, SkySat, IKONOS, or GaoFen. The free and common satellite images of Sentinel and Landsat products also have a spatial resolution between 10 – 100  m). High spatial resolution modeling approaches often go to 10 m resolution. The fine scale variability can give an estimation of the internal variability at pixel scales for such applications. However, the spatial resolution of satellite images and computational power for numerical models is in a continuous improvement, and sooner or later more products will reach the resolution of our dataset.

Finally, an important potential application of this fine scale database of GST is for simulations of the energy exchange in this dynamic environment at the land and atmosphere boundary. The 1D simulations of the energy fluxes in the process-based models will benefit from this dataset both for validation and for parameterization. These models could help to better understand this fine scale variability of GST under the influence of landcover.

GST is important not only for characterizing periglacial processes and permafrost and seasonally frozen ground evolution but could be useful for various geosciences and economic applications. For example, soil science still relies on the air temperature as input while GST is more adequate for microbiology studies and ecosystem monitoring. Even for modeling soil temperature and permafrost distribution the air temperature and land surface temperature are often used, while studies have shown the large surface offsets with GST. In precision agriculture, the GST can be used for scheduling irrigations, combating droughts and freezing for a sustainable development and management of resources.

5. The intra-plot differences at most sites are usually larger during the freeze-thaw transition period (Figs. 3-6), but at site B6 and B7, the same pattern is

not observed and the differences are large throughout the entire year (Fig.

5). What are the possible reasons? Is there anything special about these

two sites?

Now that we uniformized the Y scale with the same temperature range, for these two sites we can see that the daily differences are still reaching the maximum values during the freeze-thaw transition period. However, for the other periods of the year, even though the differences are lower they still show a spiky pattern. We believe now that the fine scale variability of the water content in the shallow soil could explain this pattern. This might be more relevant for site B6 placed in swamp meadow where even the surface water was present in a patch pattern. However, the water content can also show high variability in the bare ground sites. For example, the water content was one of the lowest in plot B7B (8%) compared to other plots from the bare ground sites that reached even 44% (Table 3). Therefore, as we stated in the reply to the first comment, additional soil samples should be taken from these key sites and plots that raised more questions. Having more parameters to compare concerning the soil properties, water content, and organic matter content, could better explain the fine-scale differences in GST even under similar landcover types.

6. Table 2. It's not surprising that R or R2 values are close to 1, but the RMSE

or MAE provide more insightful information regarding GST variation at

different scales. Additionally, investigating potential relationships between

GST differences and environmental factors like elevation might be helpful.

Including a figure to visualize these relationships could enhance the clarity

of the analysis

Indeed, as also stated by the first two referees, the main aim of this data paper is just to make available new GST data for the scientific community. A detailed analysis of the controlling factors on GST variability was performed in our previous work. The control of elevation on GST spatial variability has been assessed in detail by statistical tests and including a graphical representation of the decrease of GST with elevation (please see Figures 4 and 8 from Șerban et al., 2023). In this data paper, we avoided repeating the same analysis and figures and we focused more on the intra-plot variability of GST. We only briefly mentioned:

L83-84: "The variability of MAGST at other scales and their environmental controls have been assessed in detail by Șerban et al. (2023)."

L323-324: "The intra-site MAGST variability has been mainly controlled by elevation and landcover types ( as is shown in Figs. 4 and 8 of Șerban et al., 2023), similar to observation from the Swiss Alps (Gubler et al., 2011)."

**Technical corrections:**

Figure 1: add the lat/lons infromation, and adding a permafrost map as the

background may be also helpful.

The lat/long coordinates have been added as suggested on the inset map that shows the study area in the south-central Headwater Area of Yellow River (HAYR). On the same map, in the background, it is already added the permafrost distribution after Wang et al., (2005) as described in the caption. The permafrost distribution layer is represented with dots and a brown boundary and described in the legend as "Plateau discontinuous permafrost".

line 119-121: are these data from site CLP-1 or CLP-2?

These data are from borehole CLP-2 because that is the deep borehole of 100 m in depth, while CLP-1 is 20 m in depth. More details are in Luo et al., 2018b.

Line 156: "Photographs were taken at each site and plot". I would suggest the

authors add some photos to better present sites condition.

Photographs illustrating the site and plot conditions have been added as suggested.

The following changes have been made in the manuscript:

L163-166: "Figure 3. Photographs presenting the monitoring plots of GST in different landcover types: alpine steppe and bare ground – B1 (a); earth hummocks in alpine swamp meadow – A8 (b); fine bare ground – A4 (c); coarse bare ground – D1 (d); fine bare ground in the depression of a drained thermokarst pound and in the nearby alpine meadow – A2 (e); alpine meadow – B4 (f)." has been added.

Line178: please briefly describe what AIC is.

L195-196: "The AIC is a statistical test used to assess how well the model fits the data (Akaike, 1974)." has been added.

Line 222: change "both" to "these two"

L234-235: "At both sites, the plots are situated in a …" has been replaced with "At these two sites, the plots are situated in a ...".

Figure 3-7: I would suggest the authors using same Y scale to better show the

differences.

A Y scale ranging from –4 to 4 °C has been used for all the plots in Figures 3-6 (now Figures 4-7), while a Y scale ranging from –15 to 15 °C has been used for all plots in Figure 7 (now Figure 8).

Figure 7d: the color difference between the two lines is minimal, making it

difficult to distinguish the line representing "steppe."

The color of the steppe has been changed from yellow to purple to better be distinguished from the orange of the bare ground.

Figure 8: I would suggest sorting the sites in transect by elevation to better

present if there are elevation effects.

The sites in the transect have been sorted by elevation in both Figures 8 and 9 (now Figures 9 and 11).

**References**

Akaike, H.: A new look at the statistical model identification, IEEE Trans. Automat. Contr., 19, 716–723, https://doi.org/10.1109/TAC.1974.1100705, 1974.

Gubler, S., Fiddes, J., Keller, M., and Gruber, S.: Scale-dependent measurement and analysis of ground surface temperature variability in alpine terrain, Cryosphere, 5, 431–443, https://doi.org/10.5194/tc-5-431-2011, 2011.

Luo, D., Jin, H., Jin, X., He, R., Li, X., Muskett, R. R., Marchenko, S. S., and Romanovsky, V. E.: Elevation-dependent thermal regime and dynamics of frozen ground in the Bayan Har Mountains, northeastern Qinghai-Tibet Plateau, southwest China, Permafr. Periglac. Process., 29, 257–270, https://doi.org/10.1002/ppp.1988, 2018b.

Șerban, R. D., Bertoldi, G., Jin, H., Șerban, M., Luo, D., and Li, X.: Spatial variations in ground surface temperature at various scales on the northeastern Qinghai-Tibet Plateau, China, Catena, 222, 106811, https://doi.org/10.1016/j.catena.2022.106811, 2023.

---

## Referee Report (RR1)

The authors have meticulously revised their manuscript in response to the feedback provided during the first round of review. Their responses to the reviewer comments are detailed and rigorous, resulting in a more accurate and comprehensive presentation of the data acquisition process, the data analysis, and the application prospects. The authors have successfully addressed my concerns raised in the initial review. However, there is still room for improvement in the figures of the manuscript. To enhance the quality of the illustrations, it is recommended that the authors consider the following suggestions:

(1) Increase the font size in the figures, including both the legends and the text within the images. For instance, the map in the upper right corner of Figure 1 is challenging to decipher without special enlargement, especially regarding the place names. Figures 4 to 7 belong to the same type, yet the font sizes in the legends are inconsistent and generally too small to be clearly read.

(2) Adjust the proportions of the figures and enhance their resolution. For example, there is considerable empty space on both sides of Figure 2, which could be utilized to better showcase relevant details by appropriately expanding the image. In Figure 3, the photographs taken by the camera appear somewhat blurred, and the white annotations could be replaced with more prominent colors for better visibility.

If the authors address these graphical concerns and further refine the figures, I recommend the publication of their manuscript in ESSD.

---

## Author Response (AR2)

The authors would like to thank the editor and reviewers for their constructive feedback, and the thorough assessment of the manuscript. Below, we are providing a point-to-point response to each comment: Reviewers' comments are given in black, and our responses are given in blue. Additionally, we have included details of how we address these changes in the revised submission.

**Response to Editor**

Thank you for submitting the revision! Two reviewers believed that it has addressed the concerns and suggestions from the previous review, and the manuscript can be accepted for publication. One reviewer made good suggestions regarding the figures. I would encourage the authors to modify the mentioned figures accordingly to provide better visual effects in the final paper.

We further improved the figures according to the reviewer suggestions.

**Response to Referee 1**

The authors have meticulously revised their manuscript in response to the feedback provided during the first round of review. Their responses to the reviewer comments are detailed and rigorous, resulting in a more accurate and comprehensive presentation of the data acquisition process, the data analysis, and the application prospects. The authors have successfully addressed my concerns raised in the initial review. However, there is still room for improvement in the figures of the manuscript. To enhance the quality of the illustrations, it is recommended that the authors consider the following suggestions:

Thank you for your appreciation! We have tried our best to further improve the figures as follows.

(1) Increase the font size in the figures, including both the legends and the text within the images. For instance, the map in the upper right corner of Figure 1 is challenging to decipher without special enlargement, especially regarding the place names. Figures 4 to 7 belong to the same type, yet the font sizes in the legends are inconsistent and generally too small to be clearly read.

In Figure 1 in the upper inset, the font size has been increased from 7-8 to 10 pt.

In Figures 4 to 7 the font size of the legend has been increased to 10 pt..

(2) Adjust the proportions of the figures and enhance their resolution. For example, there is considerable empty space on both sides of Figure 2, which could be utilized to better showcase relevant details by appropriately expanding the image. In Figure 3, the photographs taken by the camera appear somewhat blurred, and the white annotations could be replaced with more prominent colors for better visibility.

If the authors address these graphical concerns and further refine the figures, I recommend the publication of their manuscript in ESSD

The empty space on both sides of Figure 2 is actually outside the figure. The two borderlines of sub-figures a) and b) delineate the margins of the Figure 2. Because the figure is centered and the caption is longer than the figure, this gives the feeling that the figure is as big as the caption and has empty spaces on both sides. To avoid this feeling a general borderline has been added to the figure.

In Figure 3 the image quality has been improved from 96 DPI to 300 DPI and the white annotations have been changed to yellow which improved the visibility as suggested.